# Convex Potential Mirror Langevin Algorithm for Efficient Sampling of Energy-Based Models

**Zitao Yang[1], Amin Ullah[2], Shuai Li[3], Li Fuxin[4], Jun Li[1]**
[1]Fudan University, [2]Boeing Research and Technology
[3]University of Bonn, [4]Oregon State University
yangzt21@m.fudan.edu.cn, amin.ullah@boeing.com, lishuai@iai.uni-bonn.de
lif@oregonstate.edu, jun_li@fudan.edu.cn

## Abstract

This paper introduces the Convex Potential Mirror Langevin Algorithm (CPMLA), a novel method to improve sampling efficiency for Energy-Based Models (EBMs). CPMLA uses mirror Langevin dynamics with a convex potential flow as a dynamic mirror map for EBM sampling. This dynamic mirror map enables targeted geometric exploration on the data manifold, accelerating convergence to the target distribution. Theoretical analysis proves that CPMLA achieves exponential convergence with vanishing bias under relaxed log-concave conditions, supporting its efficiency in adapting to complex data distributions. Experiments on benchmarks like CIFAR-10, SVHN, and CelebA demonstrate CPMLA's improved sampling quality and inference efficiency over existing techniques.

## 1 Introduction

Energy-based models (EBMs) represent a class of generative machine learning models designed to capture and synthesize complex data distributions. EBMs define an unnormalized probability distribution via an energy function, assigning low energy values to likely data samples (corresponding to the target distribution) and high energy values to unlikely ones [55, 39, 10]. Known for their conceptual simplicity and training stability, EBMs have found diverse applications ranging from 3D object recognition [15] and analysis [56] to image segmentation [24], super-resolution restoration [60], machine translation [47], and protein folding [48, 52].

A critical limitation of energy-based models (EBMs) lies in their reliance on Markov Chain Monte Carlo (MCMC) sampling methods, particularly when operating in high-dimensional data spaces [4, 10, 29]. MCMC algorithms like Langevin dynamics often get trapped in local energy minima when the underlying data manifold is characterized by multi-modal energy landscapes [21] or exhibits non-Euclidean geometry [59, 21]. When sampling from the complex, highly multi-modal energy landscapes characteristic of deep EBMs, these MCMC methods can become computationally intensive and yield biased sampling [58]. These factors hinder the efficient approximation of complex distributions and can lead to slow convergence towards the target distribution.

Recent methods address sampling inefficiencies within EBMs. Some strategies refine MCMC initialization [18, 10], while others explore gradient approximation techniques [25, 29]. However, persistent challenges such as non-mixing issues remain unresolved [58]. Mirror Langevin algorithms have recently emerged as an alternative approach to alter sampling geometry via a fixed mirror map, i.e., a predefined function. Prior work [1, 30] demonstrates that mirror Langevin algorithms, under certain assumptions, exhibit vanishing bias (bias $\rightarrow 0$ as the step size $h \rightarrow 0$). This property ensures reliable convergence to the target distribution and improves sampling accuracy. Moreover, mirror Langevin algorithms achieve mixing times independent of the domain's condition number, enabling fast convergence [19, 59]. However, fixed mirror maps in conventional mirror Langevin algorithms

struggle to capture complex data manifolds efficiently, limiting their use for large-scale problems, especially those associated with deep neural networks.

This paper introduces Convex Potential Mirror Langevin Algorithm (CPMLA), a novel approach for sampling EBMs with enhanced efficiency. Unlike conventional mirror Langevin algorithms in Euclidean space, CPMLA employs a learnable, data-driven mirror map that actively infers the intrinsic manifold structure of the data. By parameterizing the mirror map as the gradient of a convex potential function (cf. Brenier's theorem [44]), CPMLA dynamically reorients sampling trajectories to align with the non-Euclidean geometry of the target distribution, enabling adaptive exploration of high-density regions while avoiding metastable states.

We employ a cooperative learning strategy that jointly trains the dynamic mirror map and the EBM. First, the dynamic mirror map is learned by optimizing a convex potential flow (CP-Flow) [20]. Building on Brenier's theorem for optimal transport [44], this formulation guarantees that CP-Flow – defined as the gradient of a convex potential function [3] – inherently captures the geometric structure of the data distribution. Then, the EBM is trained by contrasting the energy of real samples with that of those synthesized via CPMLA. Concurrently, synthesized samples are fed back into the CP-Flow training phase. This alternating process aligns the CP-Flow's transport dynamics with the EBM's energy-based density estimation, mitigating sampling bias and accelerating sampling convergence.

We theoretically analyze the convergence of CPMLA. Based on the recent study [21], we prove exponential convergence under relaxed log-concavity assumptions with two improvements. First, we specialize our proof for the dynamic mirror map modeled with deep neural networks. Second, beyond the sampling algorithm's error, our analysis also incorporates the approximation errors from modeling both the CP-Flow and the EBM with deep neural networks. These improvements broaden the applicability to a wider range of target distributions in various machine learning tasks. To the best of our knowledge, this is the first analysis of mirror Langevin algorithms within the framework of deep neural networks, resulting in exponential convergence with vanishing bias (Theorem 4.5).

We evaluate CPMLA across several benchmark datasets, including CIFAR-10, SVHN, and CelebA. The results demonstrate that CPMLA not only achieves superior sampling quality but also exhibits enhanced inference efficiency compared to existing cooperative algorithms. Specifically, CPMLA achieves an FID score 73% lower than Flow+EBM [13, 38], indicating a substantial improvement in visual quality. Additionally, CPMLA not only achieves a lower FID score (20.85 vs. 21.16) than CoopFlow [58] with fewer inference iterations (20 vs. 30) and less time (15.92s vs. 16.84s) as shown in Table 2, but also operates with only 0.9% of the parameter count w.r.t. the flow part as shown in Table 3, underscoring its efficiency in both sampling and inference. CPMLA also excels in specialized tasks like image reconstruction and inpainting, further emphasizing its effectiveness in tackling complex image processing challenges.

Our main contributions are summarized as follows:

- We propose a novel Convex Potential Mirror Langevin Algorithm (CPMLA) for efficient sampling of EBMs. The efficiency comes from the modification of the sampling geometry through a dynamic mirror map modeled with a deep neural network.
- We provide a theoretical convergence analysis of the proposed CPMLA, specifically focusing on deep neural networks under relaxed assumptions.
- We evaluate the efficacy of our proposed algorithm through comprehensive experimental analyses on various benchmark datasets, including CIFAR-10, SVHN, and CelebA. Our experiments demonstrate that our CPMLA achieves superior sampling efficiency compared to existing methods. Furthermore, it surpasses alternative approaches in terms of sample quality and the fidelity of image reconstruction and inpainting.

## 2 Background

### 2.1 Energy-Based Models

Energy-Based Models (EBMs) characterize a probability density over data $x \in \mathbb{R}^d$ as follows:

$$p_\theta(x) = \frac{1}{Z(\theta)} \exp[f_\theta(x)] \tag{1}$$

Here, $f_\theta : \mathbb{R}^d \to \mathbb{R}$ represents the negative energy function, parameterized by a neural network with parameters $\theta$. The term $Z(\theta) = \int \exp[f_\theta(x)]dx$ is the normalizing constant, which is generally intractable to compute.

Generating samples from $p_\theta(x)$ involves Markov Chain Monte Carlo (MCMC) methods, with Langevin Monte Carlo (LMC) [50] being a prevalent choice. The LMC update rule is given by:

$$\hat{x}^{t+1} = \hat{x}^t + \frac{\delta^2}{2}\nabla_x f_\theta(\hat{x}^t) + \delta\varepsilon^t \tag{2}$$

where $\hat{x}^t$ is the sample at step $t$, $\delta$ is the step size, $\varepsilon^t \sim \mathcal{N}(0, I)$ is Gaussian noise, and the process is often initialized with $\hat{x}^0$ drawn from a simple distribution like uniform $p_0(x)$.

The parameters $\theta$ of the energy function are learned by maximizing the log-likelihood of observed data samples $x_i, i = 1, \ldots, n$ drawn from the true data distribution $p_{\text{data}}(x)$. The gradient of the log-likelihood objective is:

$$\nabla_\theta \log p_\theta(x) = \mathbb{E}_{p_{\text{data}}}\left[\nabla_\theta f_\theta(x)\right] - \mathbb{E}_{p_\theta}\left[\nabla_\theta f_\theta(x)\right] \approx \frac{1}{n}\sum_{i=1}^n \nabla_\theta f_\theta\left(x_i\right) - \frac{1}{n}\sum_{i=1}^n \nabla_\theta f_\theta\left(\hat{x}_i\right) \tag{3}$$

In this expression, $\hat{x}_i$ represents samples drawn from the current model distribution $p_\theta(x)$, usually obtained via LMC as described above. The expectation under $p_\theta$, which implicitly depends on the intractable $Z(\theta)$, is estimated using these generated samples $\hat{x}_i$. Consequently, the learning updates $\theta$ by contrasting the average gradient of the energy function evaluated on real data with the average gradient evaluated on samples generated by the model.

## 2.2 Convex Potential Flow

A foundational requirement for CPMLA to satisfy the mirror Langevin algorithm is that the dynamic mirror map must be derived from a strongly convex potential function via its gradient. To this end, we choose Convex Potential Flow (CP-Flow) [20] for this role precisely because its architecture, based on Input-Convex Neural Networks (ICNNs), guarantees this convexity property. A standard normalizing flow, in contrast, does not generally have a convex potential, making it unsuitable for a mirror Langevin framework. As shown below, CP-Flow learns a tractable probability density by approximating the optimal transport map between a noise distribution and the target data distribution, specifically minimizing the quadratic cost (Monge) problem.

**Optimal Transport** The Monge problem [49] seeks an optimal transport map $g$ minimizing the expected cost as follows:

$$J_c(p_X, p_Y) = \inf_{g:g(x)\sim p_Y} \mathbb{E}_{X\sim p_X}[c(x, g(x))] \tag{4}$$

where $c(x, y)$ is the given cost function.

**Theorem 2.1.** *(**Brenier's Theorem** [44]) Suppose $\mu$ and $\nu$ are probability measures with finite second moments, and assume that $\mu$ has a Lebesgue density $p_X$. In this case, there exists a convex potential $G$ such that the gradient map $g = \nabla G$ (uniquely defined except for a null set) provides the solution to the Monge problem in Equation 4 with square cost function $c(x, y) = ||x - y||^2$.*

To approximate the optimal solution for the Monge problem, the convex potential is modeled with several layers of an input-convex neural network (ICNN) $G_\vartheta$ [3], which is convex w.r.t the input:

$$\begin{aligned} G_\vartheta(x) &= L_{K+1}^+\left(s\left(z_K\right)\right) + L_{K+1}(x) \\ z_k &:= L_k^+\left(s\left(z_{k-1}\right)\right) + L_k(x), \quad z_1 := L_1(x) \end{aligned} \tag{5}$$

where $\vartheta$ denotes parameters of the neural network, $L(x)$ denotes a linear layer, $L^+(x)$ denotes a linear layer with positive weights, and $s$ is a non-decreasing convex activation function.

To ensure $G_\vartheta$ is strongly convex, which is required for $\nabla G_\vartheta$ to be an invertible mirror map, a quadratic term is added: $G_\alpha(x) = G_\vartheta(x) + (\alpha/2)\|x\|_2^2$. For a small positive scalar $\alpha$, this guarantees that the Hessian $\nabla^2 G_\alpha \succeq \alpha I \succ 0$. This modification ensures that the gradient $\nabla G_\alpha$ is bijective and its inverse can be computed efficiently. For brevity, we omit the subscript $\alpha$ and use $G_\vartheta$ to denote the strongly convex potential hereafter.

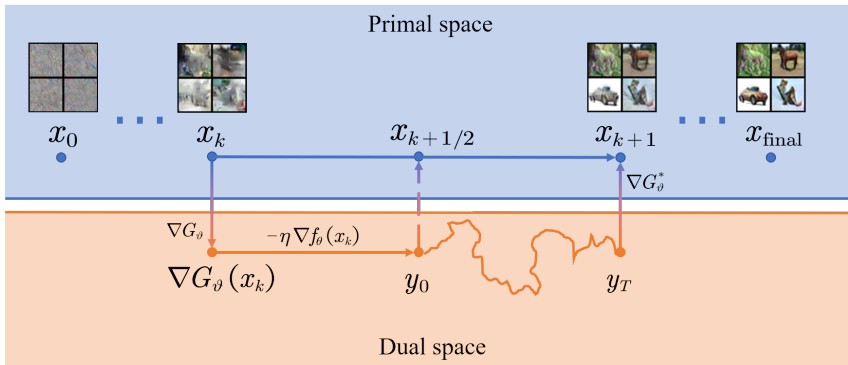

Figure 1: Overview of CPMLA sampling. Starting from a noisy sample $x_0$, CPMLA iteratively refines it by alternating between the primal space (interpretable images) and the dual space (geometry encoded by $\nabla G_\vartheta$). At each step, $x_k$ is mapped to the dual space, then updated via an EBM energy gradient step and perturbed with geometry-aware noise scaled by $\nabla^2 G_\vartheta(x_k)$. Finally, the result is mapped back to the primal space as $x_{k+1}$, yielding progressively sharper samples that efficiently explore the data manifold.

Like standard flow-based models, CP-Flow is trained by maximizing the log-likelihood of the model density. This requires computing the log-determinant of the Hessian matrix of the convex potential, $\log \det H$, where $H = \nabla^2 G_\vartheta(x)$. For high-dimensional data, forming and storing the full Hessian is computationally infeasible. To overcome this, we use a matrix-free approach based on Hutchinson's trace estimator, which relies on efficient Hessian-vector products (HVPs). The gradient of the log-determinant can be estimated as:

$$\frac{\partial}{\partial \vartheta} \log \det H = \mathbb{E}_v \left[ v^\top H^{-1} \frac{\partial H}{\partial \vartheta} v \right] \qquad (6)$$

**Algorithm 1** CP-Flow training objective

**Procedure:** Obj$(G_\vartheta, x, CG)$
Sample Rademacher $r$
**def** hvp$(v)$:
    **return** $v^\top \frac{\partial}{\partial x} \nabla G_\vartheta(x)$
$z \leftarrow$ stop_grad(CG(hvp, $r$))
**return** hvp$(z)^\top r$

where $v$ is a random vector with zero mean and unit covariance (e.g., Rademacher). The term $H^{-1}v$ is expensive to compute directly. Instead, we reframe its calculation as a quadratic optimization problem, $z^* = \arg\min_z \left\{ \frac{1}{2} z^\top H z - v^\top z \right\}$, which can be solved efficiently using the conjugate gradient (CG) algorithm without ever instantiating $H$. This procedure is summarized in Algorithm 1.

## 3 Algorithms

### 3.1 Mirror Langevin Algorithm

To generate synthesized examples from a target distribution $p(x)$ with mirror Langevin dynamics [19], we solve the stochastic differential equation:

$$\begin{aligned} dy_t &= \nabla \log p(x_t) dt + \sqrt{2 \nabla^2 G(x_t)} dW_t, \\ x_t &= \nabla G^*(y_t) \end{aligned} \qquad (7)$$

where $x_t$ and $y_t$ are stochastic processes in the primal and dual spaces, respectively, $W_t$ is the standard Brownian motion in $\mathbb{R}^d$, and $\nabla G$ is the mirror map. The term $\nabla G^*$ is the gradient of the convex conjugate $G^*$, which serves as the inverse of the mirror map, i.e., $(\nabla G)^{-1} = \nabla G^*$ (Appendix E).

We use the *Alternative Forward Discretization Scheme* (MLA$_{\text{AFD}}$) which has exponential convergence and vanishing bias [21, 1]. We use our CP-Flow $\nabla G_\vartheta$ (Section 2.2) as the dynamic mirror map. For

an iteration with step size $\eta$, the update is:

$$
\begin{aligned}
x_{k+1/2} &\stackrel{1}{=} \nabla G_\vartheta^* \left( \nabla G_\vartheta \left( x_k \right) - \eta \nabla f_\theta \left( x_k \right) \right) \\
\text{solve } dy_t &= \sqrt{2 \left[ \nabla^2 G_\vartheta^* \left( y_t \right) \right]^{-1}} dW_t \\
&\stackrel{*}{=} \sqrt{2 \nabla^2 G_\vartheta \left( \nabla G_\vartheta^* y_t \right)} dW_t \text{ for } y_0 \stackrel{2}{=} \nabla G_\vartheta \left( x_{k+1/2} \right) \\
x_{k+1} &= \nabla G_\vartheta^* \left( y_T \right)
\end{aligned}
\tag{8}
$$

The $*$ step is derived from the property of convex conjugate [2] (see Appendix F). The computation of $\nabla G_\vartheta^*$ in step 1 and $\nabla G_\vartheta$ in step 2 can be simplified by noting that they are inverses and cancel each other out in successive iterations.

## 3.2 CPMLA

Our CPMLA facilitates exploration of the underlying data manifold. It achieves this by using $\text{MLA}_{\text{AFD}}$ in Equation (8) with a CP-Flow dynamic mirror map, which dynamically transforms the sampling geometry based on the metric induced by the $\nabla^2 G_\vartheta$.

Like standard mirror Langevin methods, CPMLA alternates between updates in primal and dual spaces. The alternating sampling process entails transitioning between updating samples in the dual space using a dynamic mirror map for LMC exploration, followed by mapping the sample back to the primal space utilizing the inverse of the mirror map.

Figure 1 illustrates the sampling process of the proposed CPMLA. Specifically, each CPMLA iteration involves three steps: First, noise examples $\{y_0\}$ are generated from a standard Gaussian distribution $\mathcal{N}(0, I)$ in the dual space. And for each sampling step $k$, a noise vector $\xi_k$ is generated from a Gaussian distribution $\mathcal{N}(0, \nabla^2 G_\vartheta(x_k))$. Second, starting from $\{y_0\}$, $T$ steps of EBM sampling (gradient and SDE steps) are performed in the dual space, yielding $\{y_T\}$. Third, the inverse map transforms $\{y_T\}$ back to the primal space, yielding $\{\hat{x}\}$. The synthesized examples $\{\hat{x}\}$ are considered as outputs sampled by CPMLA.

Algorithm 2 shows the *cooperative learning* of EBM and CP-Flow. At each update, we re-initialize the MCMC chain. This is a standard practice in methods like Persistent Contrastive Divergence (PCD) to prevent chain collapse and ensure that model gradients are estimated from samples of the current model distribution, avoiding feedback from stale samples. First, we update the parameters $\vartheta$ of the CP-Flow using both original examples $\{x\}$ and synthesized examples $\{\hat{x}\}$. Then, we update the parameters $\theta$ of the EBM based on the contrast between $\{x\}$ and $\{\hat{x}\}$, as in Equation 3. The updates for both $\theta$ and $\vartheta$ are performed using the Adam optimizer, with learning rates and other hyperparameters specified in Appendix I. This cooperative mechanism simultaneously improves sampling efficiency and model expressiveness, creating a virtuous cycle of mutual enhancement.

In Algorithm 2, initial samples $\{y_0\} \sim \mathcal{N}(0, I)$ are drawn in the dual space, so no initial mapping with $\nabla G_\vartheta$ is needed. We also use a computational trick to avoid the expensive matrix square root of the Hessian: the term $\sqrt{2\eta \nabla^2 G_\vartheta(x)} \cdot \tilde{\xi}_k$ (where $\tilde{\xi}_k \sim \mathcal{N}(0, I)$) is statistically equivalent to $\sqrt{2\eta} \cdot \xi_k$ (where $\xi_k \sim \mathcal{N}(0, \nabla^2 G_\vartheta(x))$). In practice, we approximate $\nabla^2 G_\vartheta$ with its diagonal to reduce computational complexity from $O(d^3)$ to $O(d)$, enabling efficient high-dimensional sampling.

## 4 Theoretical Analysis

This section presents the convergence analysis of CPMLA, the first for mirror Langevin algorithms with deep neural network mirror maps. Our analysis relies on standard properties of neural networks (e.g., bounded gradients via clipping) and a set of theoretical assumptions, which are standard in the analysis of Langevin-type algorithms [21, 1]. We provide detailed justifications for their validity in our framework below.

**Assumption 4.1.** ($\beta$-Mirror Log-Sobolev Inequality, $\beta$-Mirror LSI) The target distribution $\pi$ satisfies $\beta$-Mirror LSI with constant w.r.t a given mirror map $\nabla G$, i.e., for every locally lipschitz function $h$, it holds that $\pi$ satisfies

$$
\frac{2}{\beta} \int \|\nabla h\|_{[\nabla^2 G]^{-1}}^2 d\pi \geq \int h^2 \log h^2 d\pi - \left( \int h^2 d\pi \right) \log \left( \int h^2 d\pi \right)
\tag{9}
$$

---

**Algorithm 2** Convex Potential Mirror Langevin Algorithm (CPMLA)

---

**Input:** (1) Observed images $\{x\} \sim p_{\text{data}}(x)$; (2) Number of Mirror Langevin steps $T$; (3) Step size in dual space $\eta$.
**Output:** Parameters of EBM and CP-Flow $\{\theta, \vartheta\}$
Randomly initialize $\theta$ and $\vartheta$.
**repeat**
    Sample noise examples $\{y_0\} \sim \mathcal{N}(0, I)$ in dual space.
    **for** $k = 0$ to $T - 1$ **do**
        Let $x_k = \nabla G_\vartheta^*(y_k)$
        Sample noise $\xi_k \sim \mathcal{N}(0, \nabla^2 G_\vartheta(x_k))$
        $y_{k+1/2} = y_k - \eta \nabla f_\theta(x_k)$
        $y_{k+1} = y_{k+1/2} + \sqrt{2\eta} \cdot \xi_k$
    **end for**
    Map back to primal space $\hat{x} = \nabla G_\vartheta^*(y_T)$
    Starting from $\{\hat{x}\}$, update $\vartheta$ by Algorithm 1
    Given $\{x\}$ and $\{\hat{x}\}$, update $\theta$ with Equation 3
**until** converged

---

**Justification**: This is a foundational assumption about the properties of the target data distribution itself, relative to the geometry induced by the mirror map. While difficult to verify empirically for complex, high-dimensional data distributions, it is a standard and necessary assumption in the literature for proving the convergence of Langevin-type algorithms in non-Euclidean spaces [21, 1]. Our contribution focuses on the aspects of the algorithm we can control and verify.

**Assumption 4.2.** ($\zeta$-Self-Concordance) There exists a constant $\zeta \geq 0$ such that the conjugate mirror map $\nabla G^*$ satisfies that $\forall y, u, s, v$,

$$\left| \nabla^3 G^*(y)[u, s, v] \right| \leq 2\zeta \cdot \left( u^\top \nabla^2 G^*(y) u \right)^{1/2} \cdot \left( s^\top \nabla^2 G^*(y) s \right)^{1/2} \cdot \left( v^\top \nabla^2 G^*(y) v \right)^{1/2} \quad (10)$$

**Justification**: This assumption bounds the third derivative of the potential function relative to its second derivative, ensuring the geometry does not change too abruptly. We empirically validate this assumption for our trained models on CIFAR-10. As direct computation of the third-order derivative tensor is infeasible, we employ a matrix-free validation approach. We estimate the Frobenius norms of the Hessian $\nabla^2 G_\vartheta(x)$ and five random directional third derivatives $\nabla^3 G_\vartheta(x)[\vec{v}]$ using Hutchinson's estimator, which relies on efficient Hessian-vector products. We then compute the proxy metric $\hat{\zeta}_{\text{proxy}} = \frac{\|\nabla^3 G_\vartheta(x)[\vec{v}]\|_F}{\|\nabla^2 G_\vartheta(x)\|_F^{1.5} + \epsilon}$. Across all training checkpoints, the value of $\hat{\zeta}_{\text{proxy}}$ consistently remains small and stable (in the range $[10^{-4}, 10^{-2}]$), providing strong empirical support that this assumption holds in practice.

**Assumption 4.3.** ($L$-Relative Lipschitz) For all $x$, it holds that $f : \mathbb{R}^d \mapsto \mathbb{R}$ is differentiable with

$$\|\nabla f(x)\|_{[\nabla^2 G(x)]^{-1}} \leq L \quad (11)$$

**Justification**: This assumption is satisfied in our framework due to standard deep learning practices. Our potential function $G$ is designed to be strongly convex, meaning its Hessian $\nabla^2 G(x) \succeq \alpha I$ for some $\alpha > 0$. In practice, we use gradient clipping on the EBM, which ensures that $\|\nabla f(x)\|$ is bounded by a constant $C$. This directly leads to $\|\nabla f(x)\|_{[\nabla^2 G(x)]^{-1}} \leq (1/\sqrt{\alpha})\|\nabla f(x)\| \leq C/\sqrt{\alpha}$. Thus, the assumption holds by setting $L = C/\sqrt{\alpha}$.

**Assumption 4.4.** (Weaker $\gamma$-Relative Smooth) For all $x, x' \in dom(G)$,

$$\|\nabla f(x) - \nabla f(x')\|_{[\nabla^2 G(x')]^{-1}} \leq \gamma \cdot \|\nabla G(x) - \nabla G(x')\|_{[\nabla^2 G(x')]^{-1}} \quad (12)$$

**Justification**: Similarly, the gradient of our EBM, $\nabla f$, is Lipschitz with some constant $L_f$ (determined by the network architecture and enforced by weight decay and gradient clipping). The potential $G$ is also smooth. This allows us to bound the relative smoothness, and the assumption holds by setting $\gamma = L_f/\alpha$.

**Theorem 4.5** (Convergence of CPMLA). *Let $d$ be the dimension of the data space. For any mirror map $\nabla G$, define $M := \exp\left(2\zeta D/\sqrt{\alpha}\right)$, where $D := \max_{u,v} \|\nabla G(u) - \nabla G(v)\|_2$ is the diameter*

*of the image of $\nabla G$. Under Assumptions 4.1-4.4, for any $\delta > 0$, after $k \geq \tilde{\Omega}\left(M\gamma^2 d/\beta^2 \delta\right)$ iterations with step size $h = O(\beta/\gamma^2 d)$, the total variation distance between the sampling distribution $\rho_t$ and the data distribution $p_{data}$ satisfies:*

$$d_{TV}(\rho_t, p_{data}) < \delta$$

*where $\tilde{\Omega}(\cdot)$ hides polylogarithmic factors, i.e. $f = \tilde{\Omega}(g) \iff \exists c > 0, n_0, p \in \mathbb{N}$ such that $f(n) \geq c \cdot \frac{g(n)}{(\log n)^p}, \forall n \geq n_0$. $\delta = \sqrt{\delta_1/2} + \delta_2 + \delta_3$, with $\delta_1, \delta_2, \delta_3$ being small constants related to the convergence error of CPMLA, approximation errors from the EBM and CP-Flow respectively.*

This theorem provides a non-asymptotic bound that characterizes the best achievable error of our framework. It states that if the EBM and the CP-Flow are trained to a certain approximation accuracy (represented by the epsilon terms), then the sampler is guaranteed to be within a certain Total Variation distance of the true data distribution. The number of iterations and step sizes are implicitly embedded in the conditions required to reach these error bounds.

**Proof sketch**: Lemma 1 from [21] provides the form of shifted drift and covariance of Equation 7 in primal space. Using this lemma, we express CPMLA in primal space as a weighted Langevin dynamics in differential form with a shifted drift term $\hat{\mu}$. Analyzing the Fokker-Planck equation for the conditional density $\rho_{t|0}(x_t \mid x_0)$, we bound the KL-divergence between $\rho_t$ and the target $\pi$ using integration by parts, the Cauchy-Schwarz inequality, and the mirror log-Sobolev inequality (Assumption 4.1). This yields a differential inequality showing exponential decay of the KL-divergence, with convergence rate governed by the algorithm's parameters and target distribution properties. The total variation bound $d_{TV}(\rho_t, p_{\text{data}}) < \delta$ decomposes into three terms: $\delta_1$ measures the distance between the distribution $\rho_k$ generated after $k$ outer iterations of CPMLA (Algorithm 2) and the stationary distribution $p_{\theta*}$ associated with the learned energy function $f_{\theta*}$. $\delta_2$ and $\delta_3$ represent the fundamental limitation in the expressive power of the chosen model architectures, namely the EBM $f_\theta$ and the CP-flow $G_\vartheta$. They reflect how well the model family can intrinsically capture the target data distribution, irrespective of sampling or optimization efficiency.

The theorem states that, incorporating slight assumptions, CPMLA not only achieves exponential convergence but also exhibits vanishing bias, making it more applicable to the practical training scenario where both the energy model and mirror map parameters are continuously updated. Details on this proof may be found in Appendix H.

## 5   Experiments

We evaluate the proposed CPMLA on diverse tasks. We start with a toy example in Section 5.1. Next, we present image generation results in Section 5.2. Finally, we demonstrate CPMLA for image reconstruction and inpainting in Section 5.3.

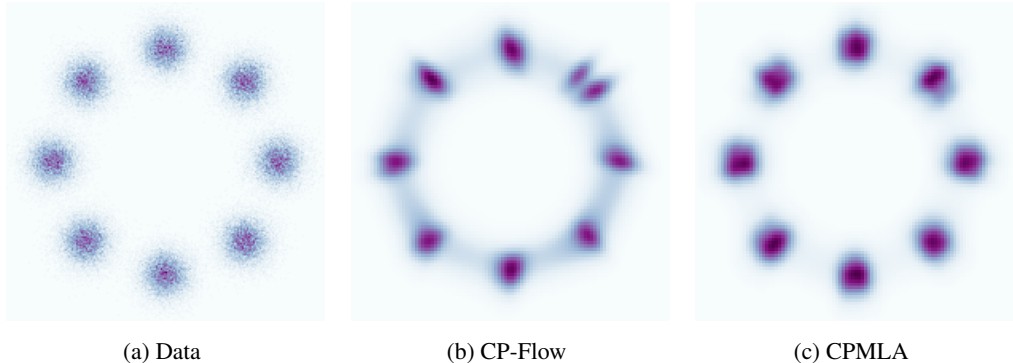

|       (a) Data       |     (b) CP-Flow      |      (c) CPMLA       |

Figure 2: Comparison between CPMLA and CP-Flow for Fitting Eight Gaussians. CPMLA reaches the same result in just 3 iterations that CP-Flow takes 10 iterations to achieve.

## 5.1 Toy Model Study

We first illustrate our approach on a toy example. Specifically, We apply CPMLA to model the eight Gaussians density from [42] and [5]. The results, presented in Figure 2, show that CPMLA efficiently fits these distributions. It demonstrates that on synthetic data, CPMLA provides a reliable approximation of the target distribution without introducing bias. Notably, CPMLA matches CP-Flow's 10-iteration result [20] in only 3 iterations, highlighting its superior convergence speed.

## 5.2 Image Generation

| Model type | Models | FID↓ |
|---|---|---|
| VAE | VAE [26] | 78.41 |
| Autoregressive | PixelCNN [43] | 65.93 |
| GAN | WGAN-GP [14] | 36.40 |
| | StyleGAN2-ADA [23] | 2.92 |
| Score-Based | NCSN [45] | 25.32 |
| | NCSN++ [46] | 2.20 |
| Flow | Glow [27] | 45.99 |
| | Residual Flow [6] | 46.37 |
| EBMs | LP-EBM [41] | 70.15 |
| | EBM-SR [39] | 44.50 |
| | EBM-IG [10] | 38.20 |
| | CoopVAEBM [57] | 36.20 |
| | CoopNets [54] | 33.61 |
| Flow+EBM | NT-EBM [38] | 78.12 |
| | EBM-FCE [13] | 37.30 |
| | CoopFlow (T=20) [58] | 30.74 |
| | CoopFlow (T=30) [58] | 21.16 |
| **CPMLA** (Ours) | CPMLAprt (T=20) | 20.85 |
| | CPMLA (T=30) | 21.09 |

Table 1: FID scores on the CIFAR-10. Our work focuses on improving the sampling efficiency and quality for the EBM family of models, making them more competitive. While other classes of generative models like score-based diffusion (e.g., NCSN++) or flow-matching models may achieve lower (better) FID scores on benchmark datasets, a direct comparison is not the primary goal. EBMs offer greater modeling flexibility, as they only require specifying an unnormalized energy function, unlike models requiring specific architectures or tractable noise processes. Our method helps make this flexibility more practical by closing the sample quality gap. The comparison to CoopFlow, a strong EBM baseline, demonstrates CPMLA's superior efficiency in this context.

We evaluate image synthesis performance on three datasets: CIFAR-10 [28], which consists of 50,000 training images and 10,000 test images across 10 categories; SVHN [37], a dataset with over 70,000 training images and more than 20,000 test images of house numbers; and CelebA [31], a large dataset of celebrity faces containing over 200,000 images. For fair comparison, all images are resized to $32 \times 32$ pixels. We present results under two settings. *CPMLA*: CP-Flow and EBM trained from scratch. *CPMLAprt*: CP-Flow is first pretrained on data, then used to initialize CPMLA training. Pretraining provides a better initialization, potentially leading to higher quality images.

We present qualitative results (Figure 3) and quantitative FID scores (Table 1). FID scores [17] are computed based on 50,000 samples. Our models outperform most baselines, achieving lower FID scores compared to standalone normalizing flows and previous EBM+flow methods [13, 38].

The results demonstrate that CPMLA is parameter-efficient and effective compared to other cooperative and flow-only approaches. In particular, compared to CoopFlow, CPMLA provides a distinct

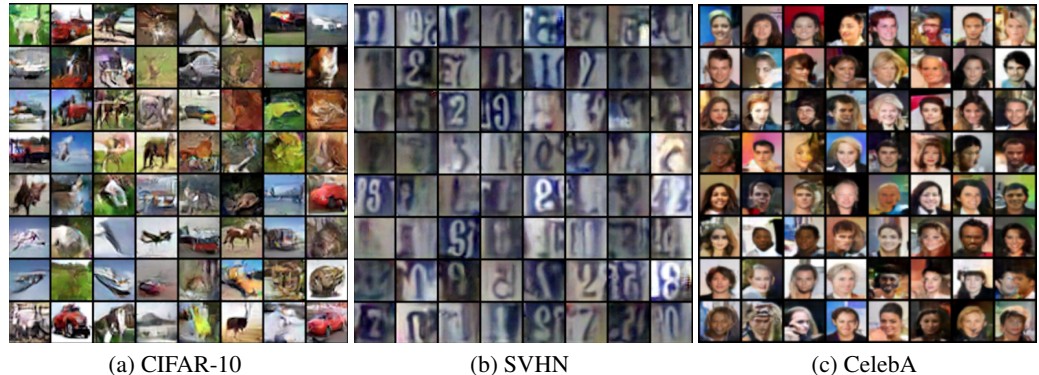

| (a) CIFAR-10 | (b) SVHN | (c) CelebA |

Figure 3: Generated Samples (32 × 32 pixels) by CPMLA from CIFAR-10, SVHN, and CelebA datasets. These images are produced under the CPMLAprt training setting.

| Models | Time (s/1k images) | FID↓ |
|---|---|---|
| CoopFlow (T=30) | 16.84 | 21.16 |
| **CPMLAprt** (T=20) | 15.92 | 20.85 |

Table 2: Wall-clock time (s/1k images) and FID comparison on CIFAR-10 (50k samples). CPMLA achieves a lower FID than CoopFlow with fewer LMC iterations and less computation time.

| | EBM part | Flow part |
|---|---|---|
| CoopFlow | 17.13M | 28.78M |
| **CPMLA** | 17.13M | 0.26M |

Table 3: Comparison of the parameter amount between CoopFlow and CPMLA. CPMLA achieves lower FID scores to CoopFlow with only 0.9% parameter count w.r.t the flow part.

advantage in terms of inference efficiency. **(i)** As shown in Table 2, **CPMLA achieves a lower FID than CoopFlow with fewer LMC iterations and less computation time**. Figure 4 further illustrates how CPMLA's FID improves faster than CoopFlow's across sampling steps ($T = 3$ to $T = 30$), highlighting its superior convergence speed. **(ii)** CPMLA's CP-Flow component uses significantly fewer parameters (0.27M) than CoopFlow's normalizing flow (28.78M, see Table 3). Remarkably, **CPMLA outperforms CoopFlow while using only 0.9% of its parameters**.

**(iii)** To further analyze computational cost, we compare the total training time of CPMLA and CoopFlow on CIFAR-10 under realistic hardware constraints. When maximizing batch size to fit within 24GB of VRAM, CoopFlow is estimated to require approximately **38 hours** for training, whereas CPMLA completes in only **10.5 hours**. While a direct per-iteration comparison for a fixed batch size shows that CPMLA is marginally slower (15.7 s/iter vs. 12.0 s/iter for CoopFlow) due to the more complex CP-Flow architecture, its memory efficiency allows for larger batches, leading to significantly better overall training throughput. This highlights CPMLA's superior training efficiency in practical, resource-constrained scenarios.

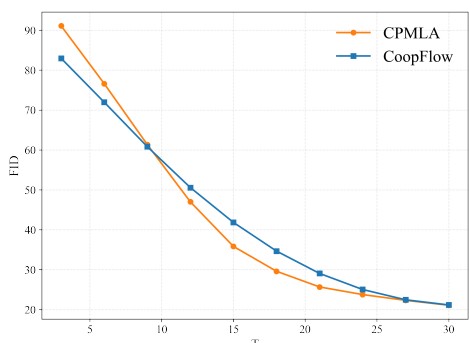

Figure 4: FID comparison from $T = 3$ to $T = 30$ between CPMLA and CoopFlow on CIFAR-10 dataset. From an inferior initialization, CPMLA demonstrates faster inference speeds than CoopFlow.

## 5.3 Image Reconstruction and Inpainting

We evaluate CPMLA for image reconstruction task, with a focus on the CIFAR-10 testing set as illustrated in Figure 6 (Appendix J). The high fidelity of reconstructions demonstrates the model's capability. This empirical evidence suggests the CPMLA framework can function effectively for reconstruction.

We further demonstrate CPMLA for image inpainting. Let's assume we have an image represented by a function $I : \Omega \subset \mathbb{R}^2 \to \mathbb{R}^3$, where $\Omega$ is the domain of the image, and $I(x, y)$ gives the color at coordinates $(x, y)$. We optimize the objective energy function Equation 13 to measure the difference between the restored region and the original image. To ensure that the restoration process does not alter the undamaged parts of the original image, we introduce a constraint: $u(x, y) = I(x, y)$ if $M(x, y) = 1$.

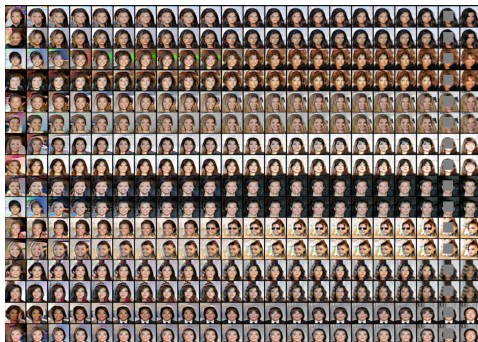

Figure 5: Image inpainting on the CelebA. The first 17 columns exhibit the inpainting results at various iterations, while the last two columns visually compare the masked images and the originals. CPMLA faithfully inpaints the masked images.

$$E(u) = \int_\Omega (I(x, y) - u(x, y))^2 \cdot M(x, y)dxdy \tag{13}$$

Experiments conducted on CelebA, are shown in Figure 5. The first 17 columns show inpainting results over optimization iterations, offering a dynamic view of the reconstruction process. The last two columns visually compare the masked images and the originals. Figure 5 shows CPMLA successfully inpaints masked images from diverse initializations.

## 6 Limitations and Future works

In our experiments, estimating the Hessian can introduce bias to the optimal point. However, compared to the exact evaluation of the inverse Hessian, this is a trade-off we must make. While our experimental results demonstrate effectiveness for diverse sampling tasks, the mirror LSI assumption (Assumption 4.1) is rather general, as we cannot ensure that the target distributions of different sampling tasks satisfy this assumption, particularly in EBMs where the target distribution is highly complex. We note that, like other generative models, improvements could potentially be misused (e.g., for deepfakes). For future work, we plan to explore the deeper connection between sampling and optimization. For instance, can optimization techniques (e.g., adaptive step sizes like Adam, trust regions) accelerate sampling or correct bias? Additionally, higher-order discretizations (e.g., Runge-Kutta) might improve convergence rates. We aim to investigate these questions and further advance the field of sampling and optimization.

## 7 Conclusions

This paper presented CPMLA, a sampling algorithm developed for Energy-Based Models (EBMs). The method utilizes Convex Potential Flow (CP-Flow) as a dynamic mirror map, allowing the sampling process to adapt to the underlying geometry of the data distribution. This adaptive mechanism facilitates sampling with vanishing bias and contributes to sampling efficiency. Theoretical analysis establishes the algorithm's convergence properties within the mirror Langevin dynamics framework. The algorithm demonstrated its applicability and effectiveness in image generation, reconstruction, and inpainting tasks. Experimental results indicated favorable performance concerning computational time and parameter count compared to related methods. In summary, CPMLA provides a principled approach to EBM sampling, integrating theoretical convergence properties with empirical performance. The method's capacity for adaptive sampling suggests its potential utility for enhancing the application of EBMs in various domains.

## Acknowledgements

Zitao Yang and Jun Li are supported by the National Natural Science Foundation of China (No. 72342016). Amin Ullah and Fuxin Li are supported by ONR/NAVSEA contracts N0014-21-1-2052 and N00024-10-D-6318.

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

# A Related Work

**Langevin sampling** Many discretizations of Langevin dynamics within Euclidean geometry have been studied in the literature, with non-asymptotic error bounds derived for various metrics like Kullback-Leibler divergence, Total Variation, and Wasserstein distance. The most extensively studied scenarios include cases where the target distribution is $m$-strongly log-concave [12, 11, 7, 9, 34] and those where it is relaxed log-concave [51, 32, 33].

**Mirror Langevin Dynamics (MLD)** extends standard Langevin dynamics by operating in a 'curved' Riemannian geometry defined by a convex potential, rather than the 'flat' Euclidean space. This allows the sampling process to adapt to the underlying geometry of the data distribution, which can lead to faster convergence. When the convex potential is quadratic, MLD reduces exactly to standard Langevin dynamics. MLD has recently gained attention in the field of non-Euclidean geometry sampling due to its superior convergence properties in constrained optimization problems. Introduced by [19] as a measure transformation of the classical Langevin dynamics, its convergence under relaxed log-concavity was investigated by [59], where the authors demonstrated convergence to a Wasserstein ball with non-vanishing bias. [1] showed vanishing bias under similar conditions as the step size decreases. [8] studied convergence using similar functional inequalities, but without exploring practical applications.

**Cooperative learning** The cooperative learning concept, first introduced in [53], involves the joint maximum likelihood training of a ConvNet-EBM [55] and a top-down generator [16]. Similarly, [57] replaced the generator in the original CoopNets with a variational autoencoder (VAE) [26] to improve inference efficiency. Our learning algorithm draws inspiration from the recent Coopflow approach [58], which collaboratively trains a Langevin flow and a normalizing flow to improve initial samples.

# B Optimal Transport

In recent years, there has been increasing interest in applying optimal transport theory to generative modeling, which considers the training process as a task of minimizing the distance between two probability distributions. More specifically, the objective is to transform a random distribution into a target distribution that closely approximates the underlying data distribution, with the distance between these two distributions often quantified using the Wasserstein distance in the context of optimal transport. The Wasserstein $p$-distance between two probability measures $\mu$ and $\nu$ on a metric space $M$ with finite $p$-moments is

$$W_p(\mu, \nu) = \left( \inf_{\gamma \in \Gamma(\mu, \nu)} \mathbf{E}_{(x,y) \sim \gamma} d(x, y)^p \right)^{1/p} \tag{14}$$

where $\Gamma(\mu, \nu)$ is the set of all couplings of $\mu$ and $\nu$. [22] shows that the Langevin dynamics manifests as the gradient flow of the Kullback-Leibler divergence within the probability measure space, characterized by the Wasserstein metric, as elucidated through the Fokker-Planck equation. This observation establishes a more substantial linkage between the realms of sampling and optimization; see also by the paper [40].

# C Mirror Langevin Algorithm for Constrained Sampling

The mirror Langevin algorithm is a powerful technique for sampling from complex distributions, particularly those with constraints or intricate geometries. It leverages the concept of mirror maps, which can adapt to the geometric structure of the target distribution, enabling efficient and accurate sampling. One notable application of the mirror Langevin algorithm is constrained sampling, where the goal is to draw samples from a population while adhering to specific conditions or constraints.

Constrained sampling involves drawing a set of samples $S$ from a population $U = u_1, u_2, ..., u_n$ while satisfying predefined constraint conditions expressed as inequalities. The general form of these constraints can be written as:

$$C(x) : g_i(x) \leq 0, \quad i = 1, 2, ..., m \tag{15}$$

Here, $g_i(x)$ represents the constraint functions, and the goal is to ensure that $g_i(x) \leq 0$ for all $i$.

The mirror Langevin algorithm addresses constrained sampling by leveraging mirror maps that can adapt to the geometry of the constraints. Specifically, we employ CP-Flow as our dynamic mirror map, which utilizes Implicit Convex Neural Networks (ICNNs) to approximate arbitrary convex functions effectively. Given that the derivative of a convex function is monotonic, we can ensure that the convergence of potential functions implies the convergence of the associated gradient fields, as stated in the following theorem:

**Theorem C.1** (Optimality (Theorem 4 in [20])). *Let $F$ be the Brenier potential of $X \sim \mu$ and $Y \sim \nu$, and let $G_n$ be a convergent sequence of differentiable, convex potentials, such that $\nabla G_n \circ X \to Y$ in distribution. Then, $\nabla G_n$ converges almost surely to $\nabla F$.*

In our context, where $\mu$ represents the unconstrained space and $\nu$ serves as the convex constraint, Theorem C.1 guarantees the existence of a convex potential whose derivative maps $\mu$ to $\nu$. By leveraging the expressive nature of ICNNs through CP-Flow, we can adapt to the intrinsic geometry of the constraints, resulting in accelerated convergence during constrained sampling. The mirror Langevin algorithm's ability to handle complex constraints makes it a valuable tool for various applications beyond constrained sampling, such as sampling from EBMs, Bayesian inference, and more [1].

# D   Derivative of CP-Flow

[20] presents an alternative formulation of the gradient as the solution to a convex optimization problem, eliminating the need to differentiate through the log-determinant estimation process. By adapting the gradient formula from Appendix C in [6] to the context of convex potentials, and utilizing Jacobi's formula* alongside the adjugate representation of the matrix inverse †, we derive the following identity for any invertible matrix $H$ parameterized by $\theta$:

$$
\frac{\partial}{\partial \theta} \log \det H = \frac{1}{\det H} \frac{\partial}{\partial \theta} \det H \overset{*}{=} \frac{1}{\det H} \operatorname{tr}\left(\operatorname{adj}(H) \frac{\partial H}{\partial \theta}\right) \overset{\dagger}{=} \operatorname{tr}\left(H^{-1} \frac{\partial H}{\partial \theta}\right) = \mathbb{E}_v\left[v^\top H^{-1} \frac{\partial H}{\partial \theta} v\right]
$$
(16)

In the last equality, [20] apply the Hutchinson trace estimator using a Rademacher random vector $v$, which is an unbiased Monte Carlo gradient estimator.

# E   Property of Convex Conjugate

$G^*$ is the convex conjugate of $G$. Then

$$
\begin{aligned}
\nabla G(x) = x^*(x) &:= \arg\sup_{x^*} \langle x, x^* \rangle - G^*(x^*) \\
\nabla G^*(x^*) = x(x^*) &:= \arg\sup_{x} \langle x, x^* \rangle - G(x)
\end{aligned}
$$
(17)

Hence

$$
x = \nabla G^*(\nabla G(x)) \quad \text{and} \quad x^* = \nabla G(\nabla G^*(x^*))
$$
(18)

# F   Lemma of Convex Conjugate

**Lemma F.1.** *Suppose we have a dualistic structure*

$$
\boldsymbol{\xi}^* = \nabla G(\boldsymbol{\xi}), \quad \boldsymbol{\xi} = \nabla G^*(\boldsymbol{\xi}^*)
$$
(19)

$G^*$ *is the Legendre dual of $G$, which is defined as*

$$
G^*(\boldsymbol{\xi}^*) = \max_{\boldsymbol{\xi}'}\left\{\boldsymbol{\xi}' \cdot \boldsymbol{\xi}^* - G(\boldsymbol{\xi}')\right\}
$$
(20)

*Then the Hessian of $G^*(\boldsymbol{\xi}^*)$ is written as*

$$
\nabla\nabla G^*(\boldsymbol{\xi}^*) = \frac{\partial \boldsymbol{\xi}}{\partial \boldsymbol{\xi}^*}
$$
(21)

*which is the inverse of the Hessian of $G(\boldsymbol{\xi})$*

$$\frac{\partial \boldsymbol{\xi}^*}{\partial \boldsymbol{\xi}} = \nabla\nabla G(\boldsymbol{\xi})$$

The last step is guaranteed by $\nabla G^* = \nabla G^{-1}$, which can be shown from Appendix E.

## G  Details of Assumptions

Previous investigations into the mirror Langevin algorithm [59] have required the relative $\mu$-strong convexity of $f$ with respect to $G$ to guarantee convergence. However, our work introduces Assumption 4.1, which relaxes this requirement and permits consideration of non-strongly convex distributions. Assumption 4.1 can be transformed as following. Taking $h(x) = \sqrt{\frac{d\rho(x)}{d\pi(x)}}$, then $\forall \rho$

$$\mathbb{D}_{KL}(\rho\|\pi) := \int \rho(x)\log\frac{\rho(x)}{\pi(x)}dx \le \frac{1}{2\beta}\int \rho(x)\left\|\nabla\log\frac{\rho(x)}{\pi(x)}\right\|_{[\nabla^2 G(x)]^{-1}}^2 dx =: \frac{1}{2\beta}J_\pi^G(\rho) \quad (22)$$

The $\mathbb{D}_{KL}(\rho\|\pi)$ term represents the KL divergence, often serving as a measure of the distance between distributions $\rho$ and $\pi$. On the other hand, the right-hand side term, $J_\pi^G(\rho)$, signifies the weighted Fisher information. As demonstrated by [22], Langevin dynamics can be interpreted as the gradient flow of the KL divergence within the space of probability measures, equipped with the Wasserstein metric through the Fokker-Planck equation. This connection establishes a link between sampling and optimization.In this context, Assumption 4.1 can be perceived as the condition of gradient domination for KL-divergence in the Wasserstein metric.

Assumption 4.2 specifically relates to the interplay between the higher-order derivatives and the lower-order derivatives of the function. When the secondary derivative is small, it implies that the first derivative, which is governed by the secondary derivative, is also small. This property ensures the solution of continuous dynamics and Hessian stability [59], indicating that the underlying geometry does not undergo rapid changes. Moreover, this property is preserved under Fenchel conjugation (with the same parameter), affine transformation and summation [36]. The concept of self-concordance is also prevalent in quadratic optimizations, such as the interior point method, where it guarantees the convergence performance $O\left(\sqrt{\zeta}\log\frac{1}{\epsilon}\right)$.

In Assumption 4.3, when $G(x) = \frac{\|x\|^2}{2}$, we regain the conventional definition of a differentiable function being Lipschitz continuous with a parameter $\beta$. This property has been extensively employed in prior research. In the case where $G = f$ and a function $f$ satisfies $\|\nabla f(x)\|_{[\nabla^2 f(x)]^{-1}} \le L$, it is referred to as a barrier function [35]. This property also emerges in the analysis of Newton's method in quadratic optimization scenarios.

In formal algorithms, it is often necessary to have the $\gamma$-relative smooth property in order to ensure convergence. $\gamma$-relative smooth is defined by

$$\begin{aligned}
&\left\|\left[\nabla^2 G(x)\right]^{-1}\nabla f(x) - \left[\nabla^2 G(x')\right]^{-1}\nabla f(x')\right\|_{\nabla^2 G(x')} \\
&\le \gamma \cdot \|\nabla G(x) - \nabla G(x')\|_{[\nabla^2 G(x')]^{-1}}
\end{aligned} \quad (23)$$

However, the CPMLA utilizes a distinct approach by employing deterministic gradient steps and stochastic steps separately. This allows for the utilization of a weaker notion of smoothness assumption, namely Assumption 4.4. Unlike the $\gamma$-relative smooth, which necessitated Lipschitz continuity across different metrics $\nabla^2 G$ and could be unavoidable when discretizing the geometry, this definition of relative smoothness only considers the local metric $\nabla^2 G$ at a single point.

## H  Proof of Theorem 4.5

*Proof.* We first clarify which parts of this proof are novel contributions and which are standard techniques adapted from prior work. Our primary contribution is the adaptation of the convergence proof of Mirror Langevin Dynamics to a setting where the mirror map is a learnable neural network

(CP-Flow) and is trained jointly with the target distribution (EBM). The overall structure of the proof, including the use of the Fokker-Planck equation and the mirror LSI, follows the framework established by [21]. Our novel steps include explicitly accounting for the approximation errors from both the EBM and CP-Flow ($\delta_2$ and $\delta_3$) and ensuring the proof holds under standard deep learning practices like gradient clipping.

For the proof, we analyze the convergence of the sampling distribution $\rho_t$ to the stationary distribution of the learned EBM, $p_{\theta^*}$. We therefore assume that $p_{\theta^*}$ satisfies the $\beta$-Mirror LSI (Assumption 4.1), as $p_{\theta^*}$ is trained to be a close approximation of the target data distribution $\pi = p_{\text{data}}$.

We decompose the total variation distance using the triangle inequality:

$$d_{TV}(\rho_t, p_{\text{data}}) \leq d_{TV}(\rho_t, p_{\theta^*}) + d_{TV}(p_{\theta^*}, q_{\vartheta^*}) + d_{TV}(q_{\vartheta^*}, p_{\text{data}}). \tag{24}$$

where $p_{\theta^*}$ is the stationary distribution of the energy model and $q_{\vartheta^*}$ is the optimal CP-Flow distribution.

For the first term, following Lemma 1 in [21], we analyze the differential form in primal space. Lemma 1 in [21] tells us that the differential form of Algorithm 2 in primal space is

$$
\begin{aligned}
dX_t = &- \left[\nabla^2 G\left(X_t\right)\right]^{-1} \nabla f\left(X_0\right) dt - \left[\nabla^2 G\left(X_t\right)\right]^{-1} \text{Tr}\left(\nabla^3 G\left(X_t\right)\left[\nabla^2 G\left(X_t\right)\right]^{-1}\right) dt \\
&+ \sqrt{2\left[\nabla^2 G\left(X_t\right)\right]^{-1}} dW_t \\
= &\left[- \left[\nabla^2 G(x_t)\right]^{-1} \nabla f(x_0) + \left[\nabla^2 G(x_t)\right]^{-1} \nabla f(x_t) - \left[\nabla^2 G(x_t)\right]^{-1} \nabla f(x_t)\right. \\
&\left. - \left[\nabla^2 G\left(X_t\right)\right]^{-1} \text{Tr}\left(\nabla^3 G\left(X_t\right)\left[\nabla^2 G\left(X_t\right)\right]^{-1}\right)\right] dt + \sqrt{2\left[\nabla^2 G\left(X_t\right)\right]^{-1}} dW_t \\
= &\left(\nabla \cdot H^{-1}\left(X_t\right) - H^{-1}\left(X_t\right) \nabla f\left(X_t\right) + \hat{\mu}\right) dt + \sqrt{2H^{-1}\left(X_t\right)} dW_t
\end{aligned}
\tag{25}
$$

where we denote $\hat{\mu} = \left[\nabla^2 G\left(X_t\right)\right]^{-1}\left(\nabla f\left(X_t\right) - \nabla f\left(X_0\right)\right)$ and $H^{-1} = \left[\nabla^2 G\right]^{-1}$.

This is a weighted Langevin dynamics with shifted drift $\hat{\mu}$ (the reason of the convergence to a biased limit).

Now consider the Fokker-Planck equation for the conditional density $\rho_{t|0}\left(x_t \mid x_0\right)$. For the drift $b = \nabla \cdot H^{-1} - H^{-1}\nabla f + \hat{\mu}$, applying Lemma 3 in [51], we have

$$
\begin{aligned}
\frac{\partial \rho_t(x)}{\partial t} &= \int \frac{\partial \rho_{t|0}\left(x \mid x_0\right)}{\partial t} \rho_0\left(x_0\right) dx_0 \\
&= \int \left[-\nabla \cdot \left(\rho_{t|0}\left(\nabla \cdot G_0(x) - G_0(x)\nabla f(x)\right)\right) + \left\langle\nabla^2, \rho_{t|0} G_0(x)\right\rangle - \nabla \cdot \left(\rho_{t|0}\hat{\mu}_0(x)\right)\right] \rho_0\left(x_0\right) dx_0 \\
&= \nabla \cdot \left(\rho_{0|t} \int -\left(\rho_t\left(\nabla \cdot G_0(x) - G_0(x)\nabla f(x)\right)\right) + \nabla \cdot \left(\rho_t G_0(x)\right) dx_0\right) - \nabla \cdot \left(\rho_t \int \rho_{0|t}\hat{\mu}_0(x) dx_0\right) \\
&= \nabla \cdot \left(\rho_{0|t} \int \rho_t G_0 \nabla \log \frac{\rho_t}{p_{\theta^*}(x)} dx_0\right) - \nabla \cdot \left(\rho_t \int \rho_{0|t}\hat{\mu}_0(x) dx_0\right) \\
&= \nabla \cdot \left(\rho_{0|t} \int \rho_t G_0 \nabla \log \frac{\rho_t}{p_{\theta^*}(x)} dx_0\right) - \nabla \cdot \left(\rho_t \int \rho_{0|t}\hat{\mu}_0(x) dx_0\right)
\end{aligned}
$$
$$\tag{26}$$

where the last equality is because $\nabla \log \frac{\rho}{p_{\theta^*}} = \nabla(\log \rho + f_{\theta^*})$.

Now consider the KL-divergence

$$\frac{d}{dt} \mathbb{D}_{KL}(\rho_t \| p_{\theta^*}) = \int \frac{d\rho_t}{dt} \log \frac{\rho_t}{p_{\theta^*}} dx + \int p_{\theta^*} \frac{1}{p_{\theta^*}} \frac{d\rho_t}{dt} dx = \int \frac{d\rho_t}{dt} \log \frac{\rho_t}{p_{\theta^*}} dx \tag{27}$$

According to Equation 26, we have

$$\frac{d}{dt}\mathbb{D}_{KL}(\rho_t\|p_{\theta^*}) = \int \frac{d\rho_t}{dt}\log\frac{\rho_t}{p_{\theta^*}}dx$$

$$= \int \nabla\cdot\left(\rho_{0|t}\int \rho_t G_0\nabla\log\frac{\rho_t}{p_{\theta^*}}dx_0\right)\log\frac{\rho_t}{p_{\theta^*}}dx - \int \nabla\cdot\left(\rho_t\int \rho_{0|t}\hat{\mu}_0 dx_0\right)\log\frac{\rho_t}{p_{\theta^*}}dx$$

$$= -\int \rho_{0|t}\int \rho_t\left\langle\nabla\log\frac{\rho_t}{p_{\theta^*}}G_0,\nabla\log\frac{\rho_t}{p_{\theta^*}}\right\rangle dx_0 dx + \int \rho_t\int \rho_{0|t}\left\langle\hat{\mu},\nabla\log\frac{\rho_t}{p_{\theta^*}}\right\rangle dx_0 dx$$

$$= -\mathbb{E}_{\rho_t}\left[\left\|\nabla\log\frac{\rho_t}{p_{\theta^*}}\right\|^2_{[\nabla^2 G]^{-1}}\right] + \mathbb{E}_{\rho_{0,t}}\left[\left\langle\hat{\mu},\nabla\log\frac{\rho_t}{p_{\theta^*}}\right\rangle\right]$$

$$\leq -\mathbb{E}_{\rho_t}\left[\left\|\nabla\log\frac{\rho_t}{p_{\theta^*}}\right\|^2_{[\nabla^2 G]^{-1}}\right] + \mathbb{E}_{\rho_{0,t}}\left[\|\hat{\mu}\|^2_{\nabla^2 G}\right] + \frac{1}{4}\mathbb{E}_{\rho_t}\left[\left\|\nabla\log\frac{\rho_t}{p_{\theta^*}}\right\|^2_{[\nabla^2 G]^{-1}}\right]$$

$$\leq -\frac{3\beta}{2}\mathbb{D}_{KL}(\rho_t\|p_{\theta^*}) + \mathbb{E}_{\rho_{0,t}}\left[\|\hat{\mu}\|^2_{\nabla^2 G}\right]$$

$$(28)$$

The third equality refers to the integration by parts formula $\int\langle\nabla G(x), v(x)\rangle dx = -\int G(x)\nabla\cdot v(x)dx$. The first inequality is because $x^\top y \leq \|x\|_2^2 + \frac{1}{4}\|y\|_2^2$ and the last inequality is from Mirror LSI (Assumption 4.1).

Under Assumption 4.2 - 4.4, let $M := \exp(2\zeta D/\sqrt{\alpha})$. We have

$$\mathbb{E}_{\rho_{0,t}}\left[\|\hat{\mu}\|^2_{\nabla^2 G}\right] \leq \gamma^2\cdot\mathbb{E}_{\rho_{0,t}}\left[\|\nabla G(x_t) - \nabla G(x_0)\|^2_{[\nabla^2 G(x_t)]^{-1}}\right]$$

$$= \gamma^2\cdot\mathbb{E}\left[\left\|-t\nabla f(x_0) + \sqrt{2}\int_0^t[\nabla^2 G(x_s)]^{1/2}dW_s\right\|^2_{[\nabla^2 G(x_t)]^{-1}}\right] \quad (29)$$

$$\leq 2\gamma^2 t^2\mathbb{E}\|\nabla f(x_0)\|^2_{[\nabla^2 G(x_t)]^{-1}} + 4\mathbb{E}\int_0^t\|\nabla^2 G(x_s)\|_{[\nabla^2 G(x_t)]^{-1}}ds$$

$$\leq 2\gamma^2 t^2 L^2 + 4t\gamma^2 Md$$

where the second inequality we use Itô isometry and $(a+b)^2 \leq 2(a^2+b^2)$.

Then if $0 \leq t \leq h$, we have

$$\frac{d}{dt}\mathbb{D}_{KL}(\rho_t\|p_{\theta^*}) \leq -\frac{3\beta}{2}\mathbb{D}_{KL}(\rho_t\|p_{\theta^*}) + 2\gamma^2 h^2 L^2 + 4h\gamma^2 Md \quad (30)$$

which is

$$\frac{d}{dt}\left(e^{\frac{3\beta}{2}t}\mathbb{D}_{KL}(\rho_t\|p_{\theta^*})\right) \leq e^{\frac{3\beta}{2}t}\left(2\gamma^2 h^2 L^2 + 4h\gamma^2 Md\right) \quad (31)$$

Integrate it for $0 \leq t \leq h$,

$$e^{\frac{3\beta}{2}h}\mathbb{D}_{KL}(\rho_h\|p_{\theta^*}) - \mathbb{D}_{KL}(\rho_0\|p_{\theta^*}) \leq \frac{2}{3\beta}\left(e^{\frac{3\beta h}{2}} - 1\right)\left(2\gamma^2 h^2 L^2 + 4h\gamma^2 Md\right) \quad (32)$$

Then

$$\mathbb{D}_{KL}(\rho_h\|p_{\theta^*}) \leq e^{-\frac{3\beta}{2}h}\mathbb{D}_{KL}(\rho_0\|p_{\theta^*}) + \frac{2}{3\beta}(1 - e^{-\frac{3\beta h}{2}})\left(2\gamma^2 h^2 L^2 + 4h\gamma^2 Md\right) \quad (33)$$

Iterating the recursion,

$$\mathbb{D}_{KL}(\rho_k\|p_{\theta^*}) \leq e^{-\frac{3\beta}{2}hk}\mathbb{D}_{KL}(\rho_0\|p_{\theta^*}) + \frac{2}{3\beta}\left(2\gamma^2 h^2 L^2 + 4h\gamma^2 Md\right) \quad (34)$$

Using Lemma 6 in [21] for initialization, picking the assumed stepsize, after $k \geq \tilde{\Omega}\left(M\gamma^2 d/\beta^2\delta\right)$, we have $\mathbb{D}_{KL}(\rho_t\|p_{\theta^*}) < \delta$.

Using Pinsker's inequality, we establish:

$$d_{TV}(\rho_t, p_{\theta^*}) \leq \sqrt{\frac{1}{2} D_{KL}(\rho_t | p_{\theta^*})} < \sqrt{\frac{\delta_1}{2}} \tag{35}$$

The second term, $d_{TV}(p_{\theta^*}, q_{\vartheta^*})$, represents the approximation error from running the LMC for a finite number of steps $T$ instead of running it to convergence. This term can be bounded by summing the incremental changes over the $T$ steps. Let $p_0 = p_{\theta^*}$ and $p_T = q_{\vartheta^*}$ be the distributions at the start and end of the sampling chain. By the triangle inequality for TV distance, we have $d_{TV}(p_{\theta^*}, q_{\vartheta^*}) \leq \sum_{t=1}^{T} d_{TV}(p_t, p_{t-1})$. We can analyze the single-step change using the Fokker-Planck equation:

$$\frac{\partial p_t(x)}{\partial t} = -\nabla_x \cdot \left( p_t(x) \frac{\eta^2}{2} \nabla_x f_\theta(x) \right) + \frac{\eta^2}{2} \nabla_x^2 p_t(x) \tag{36}$$

From this, we can estimate the incremental change as $d_{TV}(p_t, p_{t-1}) \leq \sqrt{\frac{1}{2} D_{KL}(p_t | p_{t-1})} \sim O(\eta)$. Summing over $T$ steps gives a total error of: $d_{TV}(p_{\theta^*}, q_{\vartheta^*}) \sim O(T\eta)$.

The third term leverages the universality property of CP-Flow (Theorem 3 in [20]). Given that the initial noise distribution is absolutely continuous with respect to the Lebesgue measure, there exists a sequence $q_{\vartheta_n}$ such that $d_{TV}(q_{\vartheta_n}, p_{\text{data}}) < \delta_3$ as $n > N$. The optimality of CP-Flow (Theorem 4 in [20]) further guarantees almost sure convergence in distribution of $q_{\vartheta_n}$ to the optimal Brenier map $q_{\vartheta^*}$, ensuring that $d_{TV}(q_{\vartheta^*}, p_{\text{data}}) < \delta_3$.

Combining these results, we conclude that $d_{TV}(\rho_t, p_{\text{data}}) < \delta = \sqrt{\frac{\delta_1}{2}} + \delta_2 + \delta_3$.

$\square$

# I    Experimental Details

| Parameter | Value |
|---|---|
| Dataset Size | 50,000 samples |
| Dynamic Mirror Map $\nabla G$ | 1 CP-Flow block |
| Depth | 20 |
| Dimh | 32 |
| $\nabla G$ Optimizer | Adam |
| $\nabla G$ Activation | Gaussian Softplus |
| $\nabla G$ Initial Learning Rate | 0.005 |
| EBM $\nabla f$ | 4 linear layers |
| $\nabla f$ Optimizer | Adam |
| $\nabla f$ Activation | Swish |
| $\nabla f$ Initial Learning Rate | 0.005 |
| Batch Size | 128 |
| Reported Results | After 3 and 10 epochs |

Table 4: Experimental setup for toy dataset

Table 4 outlines the experimental setup for an Eight Gaussian toy dataset experiment. This setup includes a dataset size of 50,000 samples, using single CP-Flow block with the Gaussian Softplus activation function for the Dynamic Mirror Map $\nabla G$. The optimizer for $\nabla G$ is Adam, with a and an initial learning rate of 0.005. The EBM $\nabla f$ comprises four linear layers with Swish activation, also utilizing the Adam optimizer, and an initial learning rate of 0.005. The batch size for this setup is 128, with reported results after 3 and 10 epochs.

Table 5 presents the experimental setup for the CIFAR-10, SVHN, and CelebA datasets. We use a multi-scale structure, involving two CP-Flow blocks, followed by invertible downsampling, and then another two CP-Flow blocks. All ICNN architectures had two hidden layers. The strong convexity

| Parameter | Value |
|-----------|-------|
| Datasets | CIFAR-10, SVHN, CelebA |
| Dynamic Mirror Map $\nabla G$ | 2 CP-Flow blocks |
| ICNN Architecture | 2 hidden layers |
| $\nabla G$ Optimizer | Adam |
| $\nabla G$ Activation | Gaussian Softplus |
| $\nabla G$ Initial Learning Rate | 5e-4 |
| $\nabla G$ Weight Decay | 5e-5 |
| Mirror Steps | 10 |
| Mirror Step Size | 1e-2 |
| EBM $\nabla f$ | 3 blocks with 3 convolutional layers |
| $\nabla f$ Optimizer | Adam |
| $\nabla f$ Activation | Swish |
| $\nabla f$ Initial Learning Rate | 5e-3 |
| Batch Size | 128 |
| Reported Results | After 200 epochs |

Table 5: Experimental setup for CIFAR-10, SVHN, and CelebA datasets

|  | CIFAR-10 | SVHN | CelebA |
|--|----------|------|--------|
| Training time (hours) | 10.5 | 12.6 | 40.2 |

Table 6: Training time on each dataset on eight 3090 GPUs

parameter $\alpha$ (Section 2.2) is set to 1e-4. For the EBM in CPMLA, a 3 blocks network is used to design the negative energy function. The following Table 6 presents training time of our model on each dataset on eight 3090 GPUs.

In the CPMLAprt setting, we first pretrain a CP-Flow on training examples, and then train a 30-step mirror Langevin sampling, whose parameters are initialized randomly, together with the pretrained CP-Flow by following Algorithm 2.

## J   More Image Generation Results

In Section 5.2, we have shown generated examples from CPMLA. In this section, we first show the examples generated by LMC on CIFAR-10. Then we compare the examples generated by the CP-Flow only and CPMLAprt in Figure 8 and 9. We can see huge difference between two algorithms and our generated examples are meaningful.

| Models | #Para | FID↓ |
|--------|-------|------|
| NT-EBM | 23.8M | 78.12 |
| EBM-FCE | 44.9M | 37.30 |
| GLOW | 44.2M | 45.99 |
| Flow++ only | 28.8M | 92.10 |
| CoopFlow | 45.9M | 21.16 |
| **CPMLA** | 17.39M | 20.85 |

Table 7: Model size vs. performance comparison (lower is better). CPMLA is lightweight yet highly effective, showcasing superior efficiency.

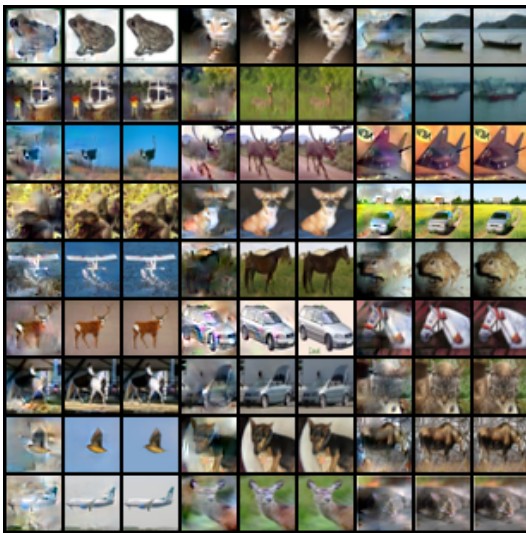

Figure 6: Image reconstruction on the CIFAR-10. The right column showcases the original images. The left and middle columns feature flow-generated images and the reconstructed images, respectively. We can see that the reconstruction is almost the same as the original one, which solidifies the stance that CPMLA functions effectively as a sampling algorithm.

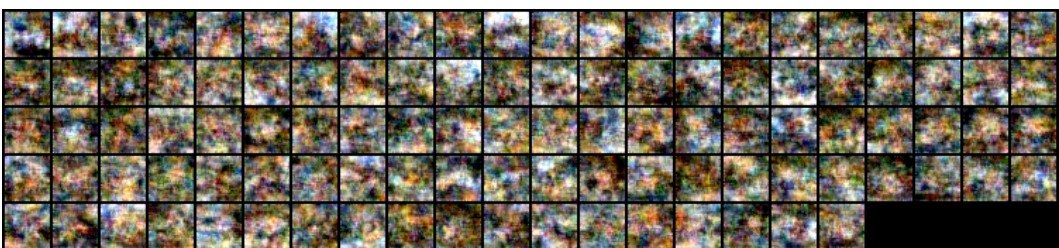

Figure 7: LMC on CIFAR-10

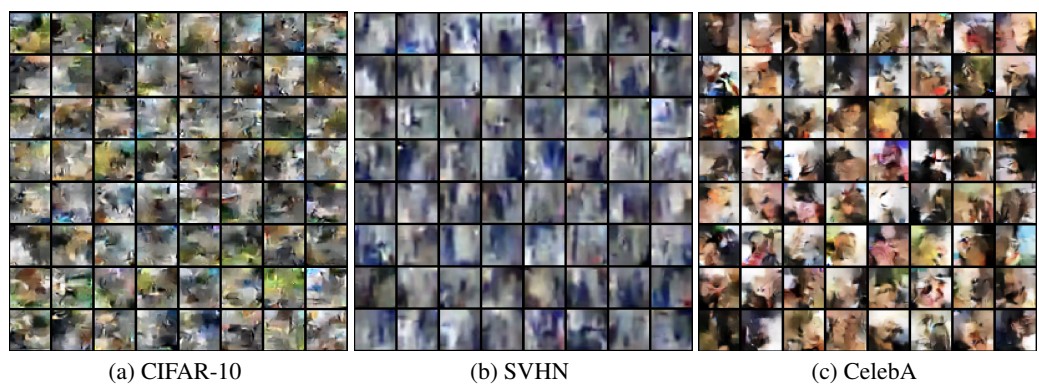

(a) CIFAR-10           (b) SVHN           (c) CelebA

Figure 8: CP-Flow results

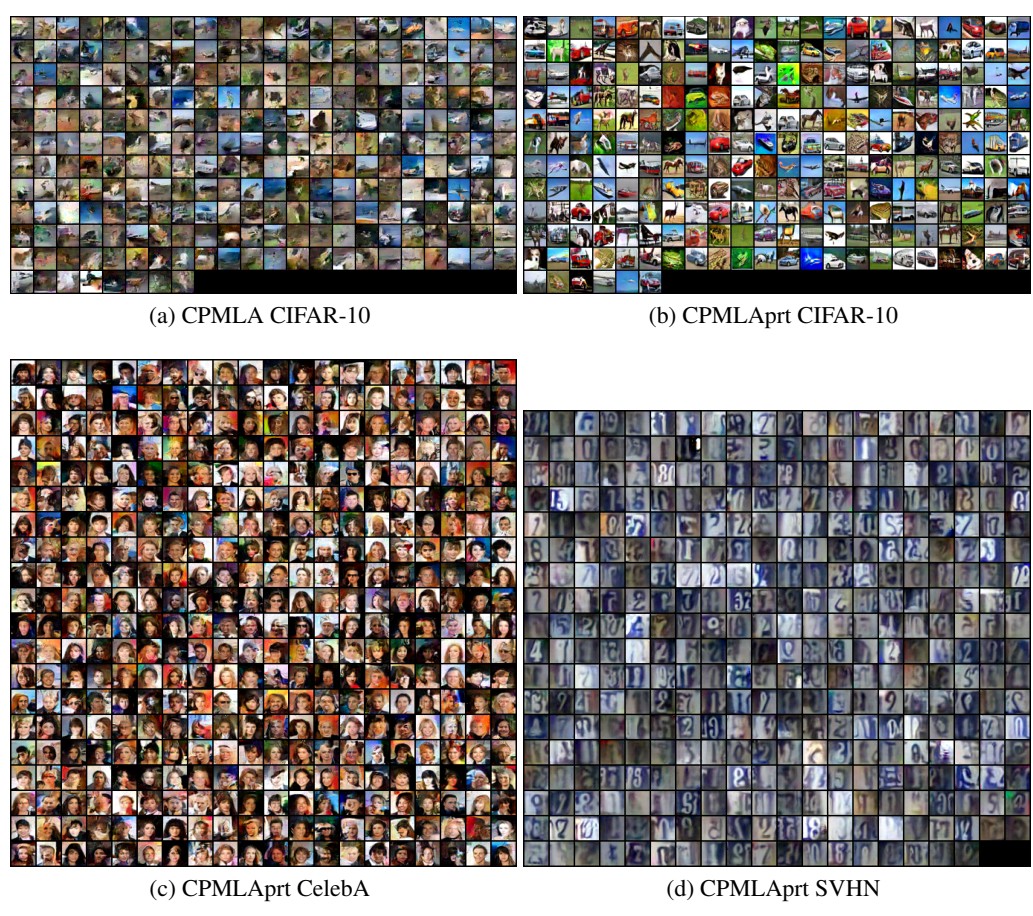

(a) CPMLA CIFAR-10

(b) CPMLAprt CIFAR-10

(c) CPMLAprt CelebA

(d) CPMLAprt SVHN

Figure 9: CPMLAprt results

