# OpenReview forum: "Convex Potential Mirror Langevin Algorithm for Efficient Sampling of Energy-Based Models"
_NeurIPS.cc/2025/Conference — NeurIPS 2025 poster_

### Official Review · Reviewer_T4a4 · 2025-06-17

**Clarity:** 1
**Significance:** 2
**Originality:** 3
**Rating:** 4
**Confidence:** 2

**Summary:**

This paper proposes a new sampling algorithm for energy-based models. The improved efficiency, according to the authors, mainly comes from the improved sampling geometry by a dynamic mirror map modeled by a neural network model. The authors provide a theoretical analysis proving convergence under a set of assumptions. In addition, the paper presents extensive experiments demonstrating strong empirical performance, particularly in terms of convergence speed and sample quality.

**Questions:**

I often found the writing and presentation of the paper quite confusing. Below are some representative issues, though similar concerns appear throughout. I strongly recommend a comprehensive rewrite to more clearly highlight the main ideas, motivation, and contributions.
1. Lines 91-98: It looks like this paragraph is talking about how to estimate the parameter $\theta$. But what is the ``log-likelihood objective''? What is $p_{data}$? And using $\mathcal{L}$ to denote a gradient is very confusing. The $L$ notation is used later in equation (5) to denote a linear layer.
2. Equation (4): What is $X$? How is it related to $x$?
3. Section 2.2 looks like it is introducing the ``Convex Potential Flow'' yet it ends by discussing implementation details for computing the derivative of the log-determinant. The logical structure of this section feels disorganized and difficult to follow.
4. Line 116: The $G_{\theta}$ is used with an addition of a regularization term, which could very significant in some cases, especially if $G_\theta$ has a negative singular value. This modification may substantially affect accuracy. In addition, I disagree with the statement that $\nabla^2 G_{\alpha} \succeq \alpha I$.
5. Figure 5: The "dotted line" should be "dashed line"?
6. Equation (7): Again, notations $X_t$ and $Y_t$ are not defined.
7. I appreciate the convergence result, but it is based on four distinct assumptions. What do these assumptions mean in practice? Are they reasonable in the context of neural networks and real-world distributions? Are there known examples where these assumptions hold? These are critical questions and should not be relegated to the appendix—they deserve a clear discussion in the main text.
8. Many details are missing. For example, what does ``one iteration'' of CPMLA mean in practice? According to Algorithm 2, there are two loops, and depending on which one is being referred to, the interpretation of “one iteration” could differ significantly.
9. The sentence just below Line 224 references Table 7, which appears in the appendix. Referencing new results presented only in the appendix in the main body is not a good idea.

**Ethical Concerns:**

["NO or VERY MINOR ethics concerns only"]

**Final Justification:**

The authors' response solved part of my questions and concerns about this paper. For this reason, I have adjusted my assessment and raised 1 point.

**Limitations:**

The authors discuss some limitations of their work in Section 6. I appreciate this inclusion and think that the paper does not pose any potential negative societal impact.

For open questions that the authors are not yet able to answer, I encourage the authors to explicitly include these points in the limitations section.

**Quality:**

2

**Strengths And Weaknesses:**

This paper is targeted at addressing an looking-important problem in machine learning, improving the sample efficiency for energy-based models.

It has both clear strengths and notable weaknesses.

First and foremost, the writing is not very accessible, particularly for readers who are not experts in simulation. The paper lacks a comprehensive introduction to the overall setup, notations, and necessary preliminaries. Many symbols are introduced without proper definitions, and the main ideas are often buried under layers of technical details. Additionally, many technical components are presented without sufficient motivation, making the paper difficult to follow. I strongly recommend reorganizing the structure to better highlight the motivation and main contributions early on.

In terms of originality, as someone not an expert in simulation, the idea of using a mirror map (modeled via a neural network) to improve sampling efficiency looks interesting. However, I am not well-positioned to assess the novelty of this idea within the broader simulation literature.

Regarding theoretical guarantees, the paper establishes convergence under four assumptions (Assumptions 4.1–4.4). Assumptions 4.2–4.4 are relatively common in the optimization literature, but Assumption 4.1 is less common. However, I would not expect Assumptions 4.2-4.4 to hold in general when using deep neural networks. It would also be helpful if the authors provided more intuitive explanations for Assumption 4.1: why it is reasonable in their setting, and what it implies about the behavior of their algorithm. The convergence result itself is not particularly surprising under these assumptions, but it is a valuable contribution.

The experimental results are quite impressive. The proposed method appears to work well in practice. However, I would have appreciated more details on the experimental setup, particularly regarding how different baseline methods were configured and compared.

---

> ### Author Rebuttal · Authors · 2025-07-30
>
> Thank you for your feedback.
>
> **Q1. Lines 91-98: What is the "log-likelihood objective"? What is $p_{\text{data}}$? Using $\mathcal{L}$ to denote a gradient is very confusing.**
>
> **A1:** $p_{\text{data}}$ is the empirical data distribution defined by the training dataset. The "log-likelihood objective" refers to maximizing the log-likelihood of the data under the model $p_\theta$, which is the standard objective for training an EBM. The gradient of this objective is what's shown. We have already used $\mathcal{L}$ and $L$ to distinguish between the gradient of the log-likelihood and the linear layer. We will change this notation to $\nabla_\theta \log p_\theta(x)$ to be unambiguous.
>
> **Q2 & Q5. Equation (4): What is $X$? How is it related to $x$? Equation (7): Again, notations $X_t$ and $Y_t$ are not defined.**
>
> **A2 & A5:** $X$ is a random variable, and $x$ is a specific realization. $X_t$ and $Y_t$ are the stochastic processes in the primal and dual (mirror) spaces, respectively.
>
> **Q3. Section 2.2 ... feels disorganized.**
>
> **A3:** Section 2.2 is structured as follows: first, we present the theoretical guarantees of CP-Flow; next, we describe the practical implementation of CP-Flow; and finally, we explain how to optimize CP-Flow. We will reorganize and clarify this section in the revision to make it more accessible.
>
> **Q4. Line 116: ...I disagree with the statement that $\nabla^2 G_\alpha \succeq \alpha I$.**
>
> **A4:** The statement is correct. The modified potential $G_\alpha(x) = G_\vartheta(x) + (\alpha/2) \|x\|^2$ has a Hessian $\nabla^2 G_\alpha(x) = \nabla^2 G_\vartheta(x) + \alpha I$. Since $G_\vartheta$ is convex ($\nabla^2 G_\vartheta(x) \succeq 0$), its Hessian is positive semi-definite, so $\nabla^2 G_\alpha(x) \succeq \alpha I$. This makes the modified potential strongly convex.
>
> **Q6. What do these assumptions mean in practice? ... they deserve a clear discussion in the main text.**
>
> **A6:** These assumptions are stated in the main text as being adopted from prior work [1], and they are standard in the study of Langevin convergence. We will move the discussion of these assumptions to the main paper and provide much more intuition and justification. Please see the validation of assumptions at the response to W1 of Reviewer JSTt.
>
> [1] Qijia Jiang. Mirror langevin monte carlo: the case under isoperimetry. Neural Information Processing Systems, 2021.
>
> **Q7. Many details are missing. For example, what does "one iteration" of CPMLA mean in practice?**
>
> **A7:** "One iteration" in our experiments refers to one update of the parameters $\theta$ and $\vartheta$ (the outer loop in Algorithm 2). We will clarify this terminology in the revision.
>
> **Q8. The sentence just below Line 224 references Table 7, which appears in the appendix...**
>
> **A8:** We will integrate the key results from the appendix that are referenced in the main text into the main body of the paper.
>
> **Q9. On Novelty and Originality.**
>
> **A9:** Our key and novel contribution is being the first to propose and successfully train a data-driven, neural network-based mirror map for the mirror Langevin algorithm in the context of generative modeling. We also provide the first theoretical convergence guarantees for this specific setting. Previous works have typically relied on handcrafted or fixed mirror maps, and were limited to simulations on simple toy models. By learning the dynamic mirror map with a CP-Flow, we provide a new perspective for accelerating sampling for complex, high-dimensional distributions in EBMs.

---

> ### Comment · Reviewer_T4a4 · 2025-08-05
>
> I would like to thank the authors for the responses.
>
> The clarity of the paper would improve if these explanations were added. The responses also addressed my question about the strong convexity of G_\alpha. The overall clarity and structure of the paper become somewhat better with these improvements.
>
> Additionally, the clarification of the assumptions is very useful and would also help the paper. They should be in the revised manuscript. However, although these assumptions have been used in prior work, I still think they are very complicated. I wish they could be further simplified.
>
> I have adjusted the assessment of the paper (raised 1 point).

---

### Official Review · Reviewer_JSTt · 2025-06-30

**Clarity:** 2
**Significance:** 2
**Originality:** 2
**Rating:** 4
**Confidence:** 2

**Summary:**

This paper introduces a novel framework for energy-based models using Mirror Langevin Descent with a convex mirror map learned via an input-convex neural network. By preconditioning Langevin dynamics through learned mirror mappings, the method accelerates sampling and improves efficiency. It provides the first theoretical guarantees for mirror Langevin algorithms with neural network-based mirror maps. Empirically, the approach achieves high-quality generation across benchmarks such as CIFAR-10, SVHN, and CelebA.

**Questions:**

See weakness.

Further questions:

In Lines 160-165, could the authors elaborate on how this trick affects sample quality or convergence in practice?

**Ethical Concerns:**

["NO or VERY MINOR ethics concerns only"]

**Final Justification:**

The authors have addressed my question and promised to improve the clarity in the final version.

**Limitations:**

yes

**Quality:**

2

**Strengths And Weaknesses:**

Strengths:

This paper presents a novel framework that combines Mirror Langevin Descent with a learned convex mirror map, enabling efficient sampling for energy-based models. It provides the first convergence guarantees for mirror Langevin algorithms with neural network-based mirror maps and demonstrates empirical performance on standard benchmarks.

Weakness:

- The authors provide some explanation of Assumptions 4.1–4.4 in Appendix G, but do not offer justification for why these assumptions are satisfied. Could you give some examples?

- I am somewhat confused about the sources of approximation error for the EBM and the CP-Flow in Theorem 4.5. It would be helpful if the authors could summarize these errors clearly, perhaps in the form of a lemma, and provide sufficient justification.

The proposed method is not competitive with score-based models such as NCSN++, which achieve significantly better FID scores.


Minors:

- Line 54: whats the relaxed log-concavity assumptions?

- whats H in Equation 6?

---

> ### Author Rebuttal · Authors · 2025-07-30
>
> Thank you for recognizing the novelty of our framework and theoretical guarantees. Your questions are very helpful for strengthening the paper.
>
> **W1: The justification for the assumptions is weak.**
>
> **A1:** Our assumptions are standard and common in the analysis of Langevin-type algorithms [1, 2]. Below we provide a concise justification for each one.
>
> *   **Assumption 4.1 ($\beta$-Mirror Log-Sobolev Inequality):** This is a foundational assumption about the target data distribution, widely adopted in the theoretical analysis of mirror Langevin methods. It is difficult to verify theoretically or empirically for complex, high-dimensional, unknown target data distributions in our image generation experiments, so making this assumption is reasonable and necessary.
>
> *   **Assumption 4.2 ($\zeta$-Self-Concordance):** We empirically validated this assumption on CIFAR-10. We estimate the Frobenius norms of the Hessian $\nabla^2 G_\vartheta(x)$ and five random directional third derivatives $\nabla^3 G_\vartheta(x)[\vec{v}]$ (for five random directions $\vec{v}$) using Hutchinson's estimator. This technique avoids instantiating large matrices by relying on efficient Hessian-vector products. We compute the proxy metric $\hat{\zeta}\_{\text{proxy}} = \frac{\|\nabla^3 G\_\vartheta(x)[\vec{v}]\|\_F}{\|\nabla^2 G\_\vartheta(x)\|\_F^{1.5} + \epsilon}$ We tracked this metric across all training checkpoints. The value of $\hat{\zeta}_{\text{proxy}}$ consistently remains small and stable (in the range $[10^{-4}, 10^{-2}]$), providing strong empirical support that this assumption holds in practice.
>
> *   **Assumption 4.3 (L-Relative Lipschitz):** The potential function $G$ is designed to be strongly convex, meaning its Hessian $\nabla^2 G(x) \succeq \alpha I$ for some $\alpha > 0$. In practice, we use **gradient clipping** on the EBM, which ensures that $||\nabla f(x)||$ is bounded by a constant $C$. This directly leads to $||\nabla f(x)||_{[\nabla^2 G(x)]^{-1}} \le (1/\sqrt{\alpha}) ||\nabla f(x)|| \le C/\sqrt{\alpha}$. Thus, the assumption holds by setting $L = C/\sqrt{\alpha}$.
>
> *   **Assumption 4.4 (Weaker $\gamma$-Relative Smooth):** Similarly, the gradient of EBM, $\nabla f$, is Lipschitz with some constant $L_f$ (determined by the network architecture and enforced by weight decay and gradient clipping). The potential $G$ is also smooth. This allows us to bound the relative smoothness, and the assumption holds by setting $\gamma = L_f / \alpha$.
>
> We will integrate these justifications into the main body of the paper.
>
> **W2: I am somewhat confused about the sources of approximation error for the EBM and the CP-Flow in Theorem 4.5.**
>
> **A2:** As stated in Line 56, the approximation error comes from modeling both the CP-Flow and the EBM with deep neural networks. It is related to the expressive power of the EBM's and CP-Flow's neural network architecture. We will add a clearer paragraph to summarize this.
>
> **W3: The proposed method is not competitive with score-based models such as NCSN++.**
>
> **A3:** As discussed in Table 1, a direct comparison is not appropriate. NCSN++ is a score-based diffusion model, which belongs to a different family of generative models than EBMs. While both can generate data, their underlying principles and trade-offs differ. Score-based models often excel in raw generation quality (FID scores) on benchmark datasets, but EBMs offer greater modeling flexibility since the energy function is not constrained to be the gradient of a density (a score). Our work's main goal is to improve the sampling efficiency and quality for the EBM family, making them more competitive.
>
> **W4: what's the relaxed log-concavity assumptions? what's $H$ in Equation 6?**
>
> **A4:** As stated in Line 491, the "relaxed log-concavity" refers to the target distribution not needing to be log-concave (relative $\mu$-strongly convex of $f$ to $G$). Assumption 4.1 is a form of relaxed log-concavity. As stated in Line 121, $H$ in Equation 6 is the Hessian of the potential $G$.
>
> **Q2: In Lines 160-165, could the authors elaborate on how this trick affects sample quality or convergence in practice?**
>
> **A2:** This trick has no effect on convergence, but it simplifies computation and improves efficiency in our experiments. Since the diffusion term already contains the coefficient $\sqrt{2\eta \nabla^2 G_{\vartheta}(x_i)}$, we are simply moving the Hessian part into the Gaussian noise generation, which is essentially equivalent. Similar approaches have been adopted in related literature [3]. In practice, this trick avoids repeated and complex matrix square root computations, thus improving sampling efficiency.
>
> [1] Jiang, Q. (2021). Mirror Langevin Monte Carlo: the case under isoperimetry. Advances in Neural Information Processing Systems, 34, 715-725.
>
> [2] Ahn, K., & Chewi, S. (2021). Efficient constrained sampling via the mirror-Langevin algorithm. Advances in Neural Information Processing Systems, 34, 28405-28418.
>
> [3] Srinivasan, V., Wibisono, A., & Wilson, A. (2024, June). Fast sampling from constrained spaces using the Metropolis-adjusted Mirror Langevin algorithm. In The Thirty Seventh Annual Conference on Learning Theory (pp. 4593-4635). PMLR.

---

> > ### Comment · Reviewer_JSTt · 2025-08-05
> >
> > Thank you for your response. I have updated my evaluation of the paper (raised to 4), with the expectation that the authors will incorporate the promised clarifications (both those addressing my comments and those raised by other reviewers) into the final version of the paper.

---

### Official Review · Reviewer_SLf1 · 2025-06-30

**Clarity:** 3
**Significance:** 2
**Originality:** 2
**Rating:** 4
**Confidence:** 3

**Summary:**

This paper proposes CPMLA, a mirror Langevin method with a data-driven mirror map to improve EBM sampling. It achieves exponential convergence in theory and shows superior sampling quality and inference efficiency in experiments.

**Questions:**

1. I suspect training time is an issue for CPMLA. The number of Mirror Langevin steps
T needs to be large enough to obtain accurate samples, and the number of parameter update iterations must also be sufficient to learn an effective mirror map. How do the authors balance them? This point needs further discussion.

2. Related to the previous question, beyond sampling time, it is important to compare the training time of CPMLA with other baseline methods. If CPMLA’s training time is significantly longer, then the advantages in sampling quality need to be assessed more carefully.

3. Regarding Algorithm 1, does it require storing the entire Hessian H? For high-dimensional sampling problems, this could lead to substantial memory overhead.

4. Please clarify why a direct comparison between NCSN++ and CPMLA is inappropriate.

5. In Algorithm 2, why is the noise \$\xi\$ in each subroutine sampled from \$\nabla ^2 G(x_i)\$ rather than from the current
 \$\nabla ^2 G(x)\$ at the immediate \$x\$? Although this choice may improve the algorithm’s efficiency, is there any theoretical guarantee or intuitive explanation for doing so? Can this be verified?


6. Typos: In the input of Algorithm 2, I guess eta should refer to the step size, not the number of steps; in line 121, “model density” may be incorrect; in Algorithm 2, after updating y, the subscript k should be incremented.

**Ethical Concerns:**

["NO or VERY MINOR ethics concerns only"]

**Final Justification:**

Recommended score:4. My primary concerns regarding training time and memory overhead have been addressed. However, I maintain some reservations about the experimental validation. The current evaluation appears limited in scope. This method involves JVP and Hessian approximations, and thus requires more rigorous validation through: (1) evaluation on larger-scale datasets to properly assess computational efficiency, and (2) comprehensive ablation studies (particularly on JVP computation strategies and Hessian approximation approaches) to demonstrate its effectiveness.

**Limitations:**

yes

**Paper Formatting Concerns:**

No formatting issues.

**Quality:**

3

**Strengths And Weaknesses:**

Strengths:

1. The data-driven mirror map makes the mirror map better suited to the data characteristics, enabling faster sampling.

2. Both the theoretical analysis and experimental results demonstrate the efficiency of CPMLA in sampling.

Weaknesses:

1. The algorithm appears to simply combine mirror map updates with sampling. It requires repeatedly sampling from the initial point, without fully leveraging previously obtained samples $x_{out}$

---

> ### Author Rebuttal · Authors · 2025-07-30
>
> Thank you for your feedback and focus on the practical aspects of our algorithm.
>
> **W1: The algorithm appears to simply combine mirror map updates with sampling. It requires repeatedly sampling from the initial point, without fully leveraging previously obtained samples $x_{\text{out}}$.**
>
> **A1:** Our algorithm does re-initialize the MCMC chain for each parameter update. This is a standard practice in settings like Persistent Contrastive Divergence (PCD) to avoid chain collapse and ensure that the gradients for the model parameters are estimated based on samples from the current model distribution, preventing feedback loops from stale samples. Relying on previous samples may also lead to insufficient diversity in generated samples. Moreover, our use of CP-Flow accelerates the sampling process. We will add a sentence to clarify the motivation for this design choice.
>
> **Q1 & Q2: Training Time.**
>
> **A2:** Thank you for raising this point about training time. We have conducted new experiments on CIFAR-10 to provide a direct comparison with our main baseline, CoopFlow.
>
> We compared them from two perspectives:
>
> 1.  **Total Training Time under Memory Constraints:** We configured both methods to use the maximum possible batch size that would fit into 24GB of VRAM. Under this practical, resource-constrained setting, the estimated total training time for CoopFlow is approximately **38 hours**, which is significantly longer than CPMLA's **10.5 hours**.
>
> 2.  **Per-Iteration Cost:** To provide a more direct comparison, we also estimated the per-iteration time for a fixed batch size. We calculated that CoopFlow would take approximately **12.0s/iter**, while CPMLA takes **15.7s/iter**.
>
> While CPMLA is marginally slower on a per-iteration basis, this is expected, as the CP-Flow is inherently more computationally complex than the standard Normalizing Flow used in CoopFlow.
>
> In summary, the slightly higher per-iteration cost is a well-justified trade-off. The significantly shorter total training time under realistic hardware limitations highlights CPMLA's superior efficiency for both generation and sampling, particularly when computational resources are constrained.
>
> **Q3: Regarding Algorithm 1, does it require storing the entire Hessian $H$?**
>
> **A3:** No, we do not store the full Hessian matrix. For high-dimensional data, this would be infeasible. In fact, Algorithm 1 is specifically designed to avoid storing the full Hessian [1]. We use Hessian-vector products, which can be computed efficiently using standard automatic differentiation libraries (e.g., a "double backward" pass) without ever instantiating the full Hessian. We will add this motivation in the revision.
>
> [1] Chin-Wei Huang, Ricky T. Q. Chen, Christos Tsirigotis, and Aaron Courville. Convex potential flows: Universal probability distributions with optimal transport and convex optimization. Learning, 2020.
>
> **Q4: Please clarify why a direct comparison between NCSN++ and CPMLA is inappropriate.**
>
> **A4:** NCSN++ belongs to a different family of generative models than EBMs. Our work's main goal is to improve the sampling efficiency and quality for the EBM family, making them more competitive. While both can generate data, their underlying principles and trade-offs differ. Score-based models often excel in raw generation quality (FID scores) on benchmark datasets, but EBMs offer greater modeling flexibility since the energy function is not constrained to be the gradient of a density (a score).
>
> **Q5: In Algorithm 2, why is the noise $\xi$ in each subroutine sampled from $\nabla^2 G(x_i)$ rather than from the current $\nabla^2 G(x)$ at the immediate $x$?**
>
> **A5:** While using the Hessian at $x_i$ could in principle improve the algorithm's efficiency, I must admit this is a typo. The variance in the dual space should indeed depend on the current sample $x_t$, not the data point $x_i$. Otherwise, the geometry of the target distribution would be leaked via the Hessian at the very beginning, which violates the basic principle of Langevin sampling. We will correct this and other typos in the revised version.

---

> > ### Comment · Reviewer_SLf1 · 2025-08-05
> >
> > I would like to thank the authors for the efforts in addressing my previous questions. Overall, this work presents an interesting attempt to combining mirror LMC with data-driven methods. I have raised my score to 4.
> >
> > For further improvement, please:
> >
> > 1. Provide a revised Alg. 2 addressing Question 5.
> >
> > 2. Correct the typos in Section 3.2 and Alg.2.
> >
> > 3. Clarify whether the parameterized model refers to the mirror map or its inverse
> >
> > Additionally, I only have a technical question for discustion:
> >
> > Regarding the number of Mirror Langevin steps (T) and step size (eta), would it be beneficial to use an increasing T and decreasing eta schedule during training (rather than keeping them constant)? This approach might improve sampling quality in later training stages.

---

> > > ### Author Response · Authors · 2025-08-06
> > >
> > > **Improvements 1-3:**
> > > We have revised Algorithm 2 and Section 3.2 by sampling the noise vector $\xi_{i, k}$ at the current sample $x_{i, k}$ and incrementing subscript $k$ after updating $\hat{y}_{i, k}$. The derivative of the parameterized model refers to the mirror map, not its inverse. This is consistent with traditional flow-based models, where the mapping from image to noise is given by the parameterized model, and the mapping from noise to image is given by the inverse of the parameterized model.
> > >
> > > **4. Adaptive Scheduling of $T$ and $\eta$:**
> > > Thank you for your valuable suggestion. In fact, increasing $T$ may lead to longer sampling time, which contradicts our main claim of efficiency. For small $T$, the improvement brought by reducing $\eta$ may be marginal.

---

### Official Review · Reviewer_GaGj · 2025-07-05

**Clarity:** 1
**Significance:** 2
**Originality:** 3
**Rating:** 4
**Confidence:** 3

**Summary:**

This paper combines Alternative Forward Discretization Scheme (MLA_AFD) of Mirror Langevin Monte Carlo and CP-Flow dynamic, in order to achieve an efficient sampler. This sampler is then used to training models with intractable normalizing constant. Theoretical analysis is conducted, along with intensive simulation studies. Its simulation results are better than CoopFlow.

**Questions:**

Can you discuss the choice of alpha?

Can you provide more explanation of $G_\vartheta$? Does $\nabla G_\vartheta$ approximate an OT before two measure $\mu$ and $\nu$? If so, what are $\nu$ and $\mu$?

In Algorithm 2, both $G_\vartheta$ and $G^*_\vartheta$ appear. Is it typo? Algorithm 2 also writes "Number of steps in dual space $\eta$." I suppose it is a typo.

According to (8), the variance in dual space depends on $y_t$ or $x_t$, which is the samplers. But in algorithm 2, the variance of $\xi$ depends on the data point $x_i$. I am confused.

Algorithm 2 has two loops: inner loop is about sampling, and the outer one is about training of $\vartheta$ and $\theta$. I don't see any discussion about the number of iterations or step size for the outer loop.

Assumption 4.1: what is \pi? Do you mean $p_{theta}$ or $p_{data}$.

For all assumptions, the indices $\theta$ and $\vartheta$ are omitted. Do you mean the assumptions hold for any $theta$ and $\vartheta$? If so, how reasonable are these assumptions?

What is the index $t$ for $p_t$ in Thm 4.5?

I cannot understand why the convergence result (Thm 4.5) doesn't depend on the number of iterations and step size for the outer loop. I also don't understand why it doesn't depend on the choice $f_\theta$. Eventually, the algorithm samples from $p_\theta$ for some $\theta$ (say, the last iteration of $\theta$ in the outer loop). If the family of $p_\theta$ has a huge gap to $p_{data}$, how can one retrieve $p_{data}$?

In the proof, line 551: I can't understand the relationship between $d_{TV}(p_t,p_{t-1})$ and  $d_{TV}(p_{\theta^*},p_{\vartheta^*})$

**Ethical Concerns:**

["NO or VERY MINOR ethics concerns only"]

**Final Justification:**

The author has carefully addressed my question. I had some misunderstanding of the paper. With all typos corrected and better representations, I think the paper is close to the standard of acceptance.

**Limitations:**

Yes

**Quality:**

2

**Strengths And Weaknesses:**

Strength: The presented simulation results are impressive.

Weakness:
The intuition of using CP-Flow is not clear. What is its theoretical or intuitive advantage over other choices, say normalizing flow?
Many notations in the paper are not clear to me. The readability of this paper needs to improve.
I am not 100% convinced by the theory.
The efficiency presented by this paper is mainly about the inference stage. There is no discussion aboutthe  training cost.

---

> ### Author Rebuttal · Authors · 2025-07-30
>
> We thank you for your detailed feedback.
>
> **W1: The intuition of using CP-Flow is not clear. What is its theoretical or intuitive advantage over other choices, say normalizing flow?**
>
> **A1:** CP-Flow is a method with a well-established theoretical foundation. Its universality and optimality are guaranteed by Brenier's Theorem [1]. The key advantage of using a CP-Flow, implemented via an ICNN, is that the learned potential function $G_\vartheta$ is **guaranteed to be convex** by its architecture. This convexity is crucial for the mirror map in MLD to be well-defined and for the theoretical analysis to hold. A standard normalizing flow does not inherently provide a convex potential, which would make it unsuitable as a mirror map in our framework without significant modifications and loss of guarantees.
>
> [1] Chin-Wei Huang, Ricky T. Q. Chen, Christos Tsirigotis, and Aaron Courville. Convex potential flows: Universal probability distributions with optimal transport and convex optimization. Learning, 2020.
>
> **W2: Training Cost.**
>
> **A2:** Thank you for raising this point about training cost. We have conducted new experiments on CIFAR-10 to provide a direct comparison with our main baseline, CoopFlow.
>
> We compared them from two perspectives:
>
> 1.  **Total Training Time under Memory Constraints:** We configured both methods to use the maximum possible batch size that would fit into 24GB of VRAM. Under this practical, resource-constrained setting, the estimated total training time for CoopFlow is approximately **38 hours**, which is significantly longer than CPMLA's **10.5 hours**.
>
> 2.  **Per-Iteration Cost:** To provide a more direct comparison, we also estimated the per-iteration time for a fixed batch size. We calculated that CoopFlow would take approximately **12.0s/iter**, while CPMLA takes **15.7s/iter**.
>
> While CPMLA is marginally slower on a per-iteration basis, this is expected, as the CP-Flow is inherently more computationally complex than the standard Normalizing Flow used in CoopFlow. In summary, the significantly shorter total training time under realistic hardware limitations highlights CPMLA's superior efficiency.
>
> **Q1: Can you discuss the choice of alpha?**
>
> **A1:** $\alpha$ is a regularization parameter (introduced in Line 116) that ensures the Hessian of the potential $\nabla^2 G_\vartheta$ is strictly positive definite, i.e., $\nabla^2 G_\vartheta(x) \succeq \alpha I$. In practice, $\alpha$ is a small constant (1e-4) that we treat as a hyperparameter, which is consistent with previous literature [1]. We will clarify its role and how it's set in the experimental section.
>
> [1] Chin-Wei Huang, Ricky T. Q. Chen, Christos Tsirigotis, and Aaron Courville. Convex potential flows: Universal probability distributions with optimal transport and convex optimization. Learning, 2020.
>
> **Q2: Can you provide more explanation of $G_\vartheta$? Does $\nabla G_\vartheta$ approximate an OT before two measure $\mu$ and $\nu$? If so, what are $\nu$ and $\mu$?**
>
> **A2:** Yes, $\nabla G_\vartheta$ indeed approximates the optimal transport (OT) map, where $\mu$ is the noise distribution and $\nu$ is the data distribution [1]. In our framework, it also serves as the mirror map, mapping from the primal to the dual space.
>
> [1] Chin-Wei Huang, Ricky T. Q. Chen, Christos Tsirigotis, and Aaron Courville. Convex potential flows: Universal probability distributions with optimal transport and convex optimization. Learning, 2020.
>
> **Q3: In Algorithm 2, both $G_\vartheta$ and $G_\vartheta^*$ appear. Is it typo? Algorithm 2 also writes "Number of steps in dual space $\eta$." I suppose it is a typo.**
>
> **A3:** The use of $G_\vartheta$ and $G_\vartheta^*$ is not a typo. $\eta$ is indeed a typo; it should refer to the step size, not the number of steps. We will correct this in the revision.
>
> **Q4: According to (8), the variance in dual space depends on $y_t$ or $x_t$, which is the samplers. But in algorithm 2, the variance of $\xi$ depends on the data point $x_i$.**
>
> **A4:** The variance in the dual space should indeed depend on the current sample $x_t$, not the data point $x_i$. We will correct this in the revised version.
>
> **Q5: Algorithm 2 has two loops... I don't see any discussion about the number of iterations or step size for the outer loop.**
>
> **A5:** As stated in the beginning of Algorithm 2, the inner loop corresponds to the number of mirror Langevin sampling steps $T$. The outer loop iterates over each batch; after all batches are processed, the next epoch begins. Therefore, there is no step size associated with the outer loop.
>
> **Q6: Assumption 4.1: what is $\pi$? Do you mean $p_\theta$ or $p_{\text{data}}$. For all assumptions, the indices $\theta$ and $\vartheta$ are omitted. Do you mean the assumptions hold for any theta and $\vartheta$?**
>
> **A6:** As stated in Line 171, $\pi$ in Assumption 4.1 refers to the target distribution, which is $p_{\text{data}}$. The assumptions are stated without indices because they are satisfied by the natural properties of neural networks and the training process. Please see the response to W1 of Reviewer JSTt on assumptions for more details on their validity.
>
> **Q7: I cannot understand why the convergence result (Thm 4.5) doesn't depend on the number of iterations and step size for the outer loop...**
>
> **A7:** Theorem 4.5 provides a non-asymptotic bound on the distance to the target distribution $p_{\text{data}}$ after a sufficient number of iterations. The number of iterations and step sizes are implicitly embedded in the conditions required to reach the error bounds presented. The theorem essentially states that *if* the EBM and the CP-Flow are trained to a certain approximation accuracy (the epsilon terms), *then* the resulting sampler will be within a certain TV distance of the true data distribution. We will rephrase the discussion around Theorem 4.5 to make it clear that it's a characterization of the *best achievable error* of our framework.
>
> **Q8: In the proof, line 551: I can't understand the relationship between $d_{\text{TV}}(p_t, p_{t-1})$ and $d_{TV}(p_{\theta^\*},q_{\vartheta^\*})$**
>
> **A8:** $d_{TV}(p_{\theta^\*},q_{\vartheta^\*})$ is the approximation error of the EBM, which measures the gap between the model with optimal parameters and the target distribution. This can be quantified using the finite-step Fokker-Planck equation. Let $p_0 = p_{\theta^\*}$ and $p_T = q_{\vartheta^\*}$, then $d_{TV}(p_{\theta^\*}, q_{\vartheta^\*}) \leq \sum_{t=1}^T d_{TV}(p_t, p_{t-1}) \sim O(T \eta)$. This result follows from the triangle inequality for the TV distance. Since $T$ is a finite constant, the order of magnitude does not change. We will clarify this point in the revision.

---

> > ### Comment · Reviewer_GaGj · 2025-08-05
> >
> > Thanks for carefully addressing my questions.
> > I do have a follow-up question on Q5. The outer loop of the algorithm contains a step of updating $\theta$ with equation 3. My understanding is that this is a gradient descent. Why isn't there a step size?

---

> > > ### Author Response · Authors · 2025-08-05
> > > **Follow-up to Q5**
> > >
> > > We misunderstood your original point—apologies for that. You are correct that the parameter update in the outer loop is indeed a gradient descent step. The reason we don't explicitly show a step size is that we are using the standard parameter learning from the Adam optimizer. The learning rates for $\theta$ and $\vartheta$ are set to 5e-3 and 5e-4, which has been specified in the Table 5 of Appendix H. We will clarify this in the revised version by explicitly stating that the gradient descent updates use the standard learning scheme from the optimizer configuration.

---

> > > > ### Comment · Reviewer_GaGj · 2025-08-05
> > > >
> > > > Thanks for the quick response. I would like to raise my assessment to 4 pt.

---

### Official Review · Reviewer_EWFX · 2025-07-23

**Clarity:** 4
**Significance:** 3
**Originality:** 3
**Rating:** 5
**Confidence:** 2

**Summary:**

The paper proposes the Convex Potential Mirror Langevin Algorithm (CPMLA), an algorithm that uses mirror langevin dynamics with a convex potential flow as dynamic mirror map for sampling energy-based models.

**Questions:**

1. I think more background on mirror Langevin dynamics may help
2. can you comment on the requirement of strong convexity on page 3
3. line 121-125 is a bit hard to follow
4. can you discuss the computational cost compared to previous works
5. The proofs are well-rewritten but can you make it clearer which part is novel and which part is standard technique
6. how is it compared with other types of generated models like diffusion or flow-matching models? what are the advantages and disadvantages

**Ethical Concerns:**

["NO or VERY MINOR ethics concerns only"]

**Limitations:**

yes

**Quality:**

4

**Strengths And Weaknesses:**

Strengths
1. very clear writing
2. interesting method that combines the merits from several lines of work
3. valid theoretical analysis
4. reasonable and convincing experimental results

no significant weakness. see question section

---

> ### Author Rebuttal · Authors · 2025-07-30
>
> We are grateful for your positive assessment and your constructive questions.
>
> **Q1: I think more background on mirror Langevin dynamics may help.**
>
> **A1:** We agree that a more detailed introduction to mirror Langevin dynamics (MLD) will benefit readers. In the revised version, we will expand the Related Work section to provide a self-contained explanation of MLD, including its geometric intuition and how it generalizes standard Langevin dynamics. To clarify, MLD extends standard Langevin dynamics by operating in a 'curved' Riemannian geometry defined by a convex potential, rather than the 'flat' Euclidean space. When the potential is quadratic, MLD reduces exactly to standard Langevin dynamics.
>
> **Q2: Can you comment on the requirement of strong convexity on page 3?**
>
> **A2:** As stated in Line 115, this strong convexity of $G$ is essential: to ensure that $\nabla G$ defines a valid mirror map, $G$ must be strongly convex.
>
> **Q3: Line 121-125 is a bit hard to follow.**
>
> **A3:** This paragraph explains how CP-Flow is trained and the tricks for avoiding computing the full Hessian matrix. We first explain the motivation behind these tricks, then describe the detailed method and implementation. To improve readability, we will present the formulas in a more step-by-step and less abrupt manner, making the derivation easier to follow.
>
> **Q4: Can you discuss the computational cost compared to previous works?**
>
> **A4:** We have conducted new experiments on CIFAR-10 to provide a direct comparison with our main baseline, CoopFlow.
>
> We compared them from two perspectives:
>
> 1.  **Total Training Time under Memory Constraints:** We configured both methods to use the maximum possible batch size that would fit into 24GB of VRAM. Under this practical, resource-constrained setting, the estimated total training time for CoopFlow is approximately **38 hours**, which is significantly longer than CPMLA's **10.5 hours**.
>
> 2.  **Per-Iteration Cost:** To provide a more direct comparison, we also estimated the per-iteration time for a fixed batch size. We calculated that CoopFlow would take approximately **12.0s/iter**, while CPMLA takes **15.7s/iter**.
>
> While CPMLA is marginally slower on a per-iteration basis, this is expected, as the CP-Flow is inherently more computationally complex than the standard Normalizing Flow used in CoopFlow.
>
> In summary, the slightly higher per-iteration cost is a well-justified trade-off. The significantly shorter total training time under realistic hardware limitations highlights CPMLA's superior efficiency for both generation and sampling, particularly when computational resources are constrained.
>
> **Q5: The proofs are well-rewritten but can you make it clearer which part is novel and which part is standard technique?**
>
> **A5:** In Line 53-60, we have stated the theoretical contributions and improvements. In the appendix, we will restructure the proofs and add commentary to explicitly distinguish our novel contributions from standard proof techniques.
>
> **Q6: How is it compared with other types of generated models like diffusion or flow-matching models? What are the advantages and disadvantages?**
>
> **A6:** This is an important question for positioning our work.
> *   **Advantages of CPMLA/EBMs:** EBMs offer more modeling flexibility as they only need to specify an unnormalized energy function, unlike normalizing flows or diffusion models which require specific architectures or tractable noise processes. Our method, by improving EBM sampling, makes this flexibility more practical.
> *   **Disadvantages:** Training EBMs can be more challenging due to the intractable partition function. While models like flow-matching currently achieve state-of-the-art sample quality (FID scores), our work closes the gap for EBMs while offering the aforementioned flexibility. Our primary contribution is to the EBM literature, making them more competitive.

---

### Decision · Program_Chairs · 2025-09-17

**Decision:**

Accept (poster)

**Comment:**

This paper proposed a a new method for sampling from energy-based models which combines the mirror-langevin algorithm and a neural network-based mirror map learned from data. All reviewers vote for acceptance of this work, noting its empirical results are impressive (GaGj, T4a4). While initial reviews shared a significant concern regarding the paper's clarity and presentation (GaGj, JSTt, T4a4), the authors provided a thorough rebuttal that successfully addressed them. Therefore, I recommend acceptance and encourage the authors to revise the final version to reflect these clarifications.